

# Quantum adiabaticity in many-body systems and almost-orthogonality in complementary subspace

## Jyong-Hao Chen[1,2]⋆ and Vadim Cheianov[1]

**1** Instituut-Lorentz, Universiteit Leiden, P.O. Box 9506, 2300 RA Leiden, The Netherlands
**2** Department of Physics, National Central University, Chungli 32001, Taiwan

⋆ jyonghaochen@gmail.com

## Abstract

We investigate why, in quantum many-body systems, the adiabatic fidelity and the overlap between the initial state and instantaneous ground states often yield nearly identical values. Our analysis suggests that this phenomenon results from an interplay between two intrinsic limits of many-body systems: the limit of small evolution parameters and the limit of large system sizes. In the former case, conventional perturbation theory provides a straightforward explanation. In the latter case, a key insight is that pairs of vectors in the Hilbert space orthogonal to the initial state tend to become nearly orthogonal as the system size increases. We illustrate these general findings with two representative models of driven many-body systems: the driven Rice-Mele model and the driven interacting Kitaev chain model.

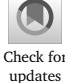

# 1    Introduction

In the modern era of quantum technologies, it is crucial to prepare and manipulate quantum states with precision. For this purpose, various approximations based on the *quantum adiabatic theorem* (QAT) [1–4] are widely used. Examples of applications include adiabatic quantum transport [5–8], adiabatic quantum computation [9–11], and adiabatic quantum state manipulation [12–17] and preparation [18–23]. Adiabatic evolution refers to the evolution of a quantum system whose time-evolved state remains close to its instantaneous eigenstate. A well-known adiabatic criterion [4, 24][1] is that the rate of change of the time-dependent Hamiltonian must be much smaller than certain non-negative powers of the minimum energy gap of the Hamiltonian. Although calculating energy gaps is often possible for systems with a small Hilbert space or for exactly solvable models, it is generically difficult for many-body systems with a large Hilbert space. Alternatively, the formalism of *shortcuts to adiabaticity* (STA) promises to give the same adiabatic results as those provided by the QAT but without requiring slow driving [25–32]. Yet, since counterdiabatic terms in many-body systems are not necessarily local in space [33–36], there are circumstances in which the STA approach is not useful for practical purposes.

Instead, a complementary bottom-up approach considers how the fidelity (termed *adiabatic fidelity*) between the time-evolved states and the instantaneous eigenstates deviates from unity under dynamical evolution. However, for quantum many-body systems, obtaining time-evolved states and instantaneous eigenstates by solving the many-body Schrödinger equation and eigenvalue equation may be a difficult task. It then raises the question of whether one can estimate the adiabatic fidelity without solving equations from scratch.

The approach initiated by Ref. [37] and extended in Ref. [38] is that the adiabatic fidelity can be estimated by exploiting a many-body nature of the problem — *generalized orthogonality catastrophe* (GOC), and a fundamental inequality for the evolution of unitary dynamics in Hilbert spaces — *quantum speed limit* (QSL). The GOC refers to the property wherein the overlap between instantaneous ground states and the initial ground state decays exponentially as both the system size and the value of the evolution parameter increase [37], whereas the QSL sets an intrinsic limit on how fast the time-evolved state can deviate from the initial state [39–45].

---

[1]See also Ref. [11] for a comprehensive review.

While the estimates derived in this manner effectively capture the behavior of the actual adiabatic fidelity within the parameter space, an unresolved question remains, first observed numerically in Ref. [37]: *Why are the numerical values of the adiabatic fidelity and the ground state overlap nearly identical in various situations, such as when the evolution parameter is small or when the system size is large?* The primary aim of this study is to address this question.

Clarifying the relation between adiabatic fidelity and ground state overlap is not only of conceptual interest but also of practical value, as it enables adiabatic fidelity to be estimated from more accessible quantities without solving the full dynamics. Our goal is therefore to provide a mechanism-level explanation for the observed closeness and to derive quantitative bounds that render this observation useful for diagnosing adiabaticity in many-body systems.

## Summary of main results and contributions

In this work, we clarify why the adiabatic fidelity and the ground state overlap often yield nearly identical values in many-body systems. We show that this closeness originates from the interplay of two complementary mechanisms (Sec. 3). First, when the driving parameter is small, perturbation theory shows that the time-evolved state remains closely aligned with the initial state, implying that adiabatic fidelity and ground state overlap coincide up to quadratic order [Eq. (24)]. By contrast, as the evolution parameter increases, perturbative arguments lose adequacy. In this regime, the explanation lies in an intrinsic property of high-dimensional Hilbert spaces: vectors within the subspace complementary to the initial state tend to become nearly orthogonal to one another. This *almost-orthogonality* (Sec. 3.2) strongly suppresses the difference between adiabatic fidelity and ground state overlap.

To substantiate these insights, we establish reverse triangle inequalities (Sec. 4) that connect adiabatic fidelity, ground state overlap, and the auxiliary overlap term [Eq. (30)], thereby providing a quantitative explanation for why adiabatic fidelity and ground state overlap remain close whenever the auxiliary overlap is small— a manifestation of the two complementary mechanisms mentioned above. We illustrate these general findings using the driven Rice–Mele model (Sec. 5), a free-fermion model amenable to partial analytical treatment; Analytic and numerical results confirm indistinguishability between adiabatic fidelity and ground state overlap already for system sizes of order $10^2$ [Eq. (33) and Fig. 2]. We further examine the robustness using the interacting Kitaev chain (Sec. 7), where exact diagonalization shows that almost-orthogonality—and thus the closeness of adiabatic fidelity and ground state overlap—persists in a nonintegrable setting.

We also propose a scaling description of the auxiliary overlap in terms of the ground-state overlap (Sec. 6) via an effective exponent $s$ [Eq. (39)]. This leads to a family of refined inequalities [Eq. (40)] for estimating adiabatic fidelity and to a modified adiabaticity breakdown criterion in which the critical driving rate acquires a finite multiplicative factor [Eq. (42)].

Taken together, these results demonstrate that the near-equality of adiabatic fidelity and ground state overlap is a robust and generic feature of driven many-body dynamics, governed by perturbative behavior at small parameters together with almost-orthogonality at large system sizes. Beyond conceptual clarification, our results provide practical diagnostics for driven many-body dynamics.

**Relation to mathematical results on many-body adiabatic state preparation.** Rigorous many-body adiabatic theorems provide *sufficient* conditions for successful adiabatic preparation, typically formulated in terms of spectral gap conditions and smooth (e.g., Gevrey-class) ramp assumptions, with driving rates that must decrease as system size increases (see, e.g., [24, 46–51]). Our approach is complementary: rather than offering sufficient condition guarantees under asymptotically slow ramps, we develop *diagnostic*, computable criteria for

adiabaticity in finite many-body systems. We likewise conclude that the driving rate must decrease with system size to maintain adiabaticity; in contrast, our scaling is inferred from the decay of adiabatic fidelity and ground state overlap rather than from spectral gap or Gevrey-class assumptions. Thus, our results provide *necessary* diagnostics directly applicable to concrete systems.

**Organization.**    The rest of this paper is organized as follows. After reviewing the basic formalism in Sec. 2, we consider a limiting case in Sec. 3 where the essential elements for addressing the proposed question are exposed. We then derive a set of useful triangle-type inequalities in Sec. 4. Our general results are illustrated using a driven Rice–Mele model in Sec. 5 and a driven interacting Kitaev chain model in Sec. 7. Implications and summary are presented in Sec. 6 and Sec. 8, respectively. The appendices provide additional technical details.

## 2 Preliminaries

### 2.1 Setup

We begin by defining notations and terminologies. Consider a time-dependent Hamiltonian $H_\lambda$ with $\lambda = \lambda(t)$ being an explicit function of time $t$. Using $\lambda$ in place of $t$ as the evolution parameter, the Schrödinger equation for the time-evolved state $|\Psi_\lambda\rangle$ reads

$$i\Gamma\partial_\lambda|\Psi_\lambda\rangle = H_\lambda|\Psi_\lambda\rangle, \quad \text{with} \quad |\Psi_0\rangle = |\Phi_0\rangle, \tag{1}$$

where $\Gamma := \partial_t\lambda(t)$ being the *driving rate* and we assume that the initial state, $|\Psi_0\rangle$, is in the ground state of the Hamiltonian $H_\lambda$ at $\lambda = 0$, $|\Phi_0\rangle$. For generic values of $\lambda$, the instantaneous ground state of the Hamiltonian $H_\lambda$ is the solution to the eigenvalue problem,

$$H_\lambda|\Phi_\lambda\rangle = E_{\text{GS},\lambda}|\Phi_\lambda\rangle, \tag{2}$$

with $E_{\text{GS},\lambda}$ being the $\lambda$-dependent ground state energy. To quantify the distance between $|\Psi_\lambda\rangle$ and $|\Phi_\lambda\rangle$, one introduces the quantum fidelity $\mathcal{F}(\lambda)$ between them,

$$\mathcal{F}(\lambda) := |\langle\Phi_\lambda|\Psi_\lambda\rangle|^2. \tag{3}$$

Let $\mathcal{C}(\lambda)$ be the overlap between the initial ground state $|\Phi_0\rangle$ and the instantaneous ground state $|\Phi_\lambda\rangle$ for an arbitrary value of $\lambda$,

$$\mathcal{C}(\lambda) := |\langle\Phi_\lambda|\Phi_0\rangle|^2. \tag{4}$$

For a large class of many-body systems, the ground state overlap $\mathcal{C}(\lambda)$ has an asymptotic form

$$\mathcal{C}(\lambda) \sim e^{-C_N\lambda^2}, \qquad C_N > 0, \tag{5}$$

under the limit of large system size, i.e. $N \to \infty$ [37]. Generalized orthogonality catastrophe, renaissance of Anderson's orthogonality catastrophe [52,53], takes place if the exponent $C_N \to \infty$ as $N \to \infty$. The scaling form of the exponent $C_N$ depends on the type of driving, space dimensions, and whether the energy gap is present or not [37].

At any value of $\lambda$, we are given three vectors: $|\Phi_\lambda\rangle$, $|\Psi_\lambda\rangle$, and $|\Phi_0\rangle$. There are three ways to construct overlaps between any two of the three vectors. We have already mentioned two kinds of the overlaps, namely, the adiabatic fidelity $\mathcal{F}(\lambda)$ (3) and the ground state overlap

$\mathcal{C}(\lambda)$ (4). The remaining overlap, $|\langle\Psi_\lambda|\Phi_0\rangle|^2$, can be utilized to define the distance between the initial state $|\Phi_0\rangle$ and the time-evolved state $|\Psi_\lambda\rangle$ through the *Bures angle* $\theta(\lambda)\in[0,\pi/2]$,

$$\theta(\lambda) := \arccos|\langle\Psi_\lambda|\Phi_0\rangle|. \tag{6}$$

Given the triplet $\{\mathcal{F}(\lambda),\mathcal{C}(\lambda),\theta(\lambda)\}$, a bound on the adiabatic fidelity $\mathcal{F}(\lambda)$ around ground state overlap $\mathcal{C}(\lambda)$ was first found in Ref. [37],

$$|\mathcal{F}(\lambda)-\mathcal{C}(\lambda)| \le \theta(\lambda). \tag{7}$$

To facilitate practical use, we employ a Mandelstam–Tamm–type quantum speed-limit inequality [39–45], which sets an upper bound on the Bures angle $\theta(\lambda)$ (6),

$$\theta(\lambda) \quad \le \quad \min\left(\mathcal{R}(\lambda),\frac{\pi}{2}\right) =: \widetilde{\mathcal{R}}(\lambda), \tag{8a}$$

$$\text{where} \quad \mathcal{R}(\lambda) := \int_0^\lambda \frac{\mathrm{d}\lambda'}{|\Gamma(\lambda')|}\sqrt{\langle H_{\lambda'}^2\rangle_0 - \langle H_{\lambda'}\rangle_0^2}, \tag{8b}$$

with $\langle\cdots\rangle_0 := \langle\Phi_0|\cdots|\Phi_0\rangle$. In this work we focus on the time-dependent Hamiltonian $H_\lambda$ of the following form:

$$H_\lambda = H_0 + \lambda V, \tag{9}$$

for which the function $\mathcal{R}(\lambda)$ (8b) with a positive constant driving rate $\Gamma$ reads

$$\mathcal{R}(\lambda) = \frac{\lambda^2}{2\Gamma}\delta V_N, \quad \text{with} \quad \delta V_N := \sqrt{\langle V^2\rangle_0 - \langle V\rangle_0^2}. \tag{10}$$

## 2.2 Orthogonal decomposition

We now reformulate the main formalism developed in Ref. [38] using a more concise projection operator approach for later use. Define $P = |\Phi_0\rangle\langle\Phi_0|$ as a projector onto the initial state and $Q = \mathbb{I} - P$ as the complementary projector. By definition, $P^2 = P, Q^2 = Q$, and $PQ = QP = 0$. Consider the following orthogonal decompositions for the time-evolved state $|\Psi_\lambda\rangle$ and the instantaneous ground state $|\Phi_\lambda\rangle$,

$$|\Psi_\lambda\rangle = P|\Psi_\lambda\rangle + Q|\Psi_\lambda\rangle, \qquad |\Phi_\lambda\rangle = P|\Phi_\lambda\rangle + Q|\Phi_\lambda\rangle. \tag{11}$$

Notice that, by the construction of Eq. (11), the following relations hold (here, $\||\cdot\rangle\| = \sqrt{\langle\cdot|\cdot\rangle}$),

$$\|P|\Psi_\lambda\rangle\| = |\langle\Phi_0|\Psi_\lambda\rangle| = \cos\theta(\lambda), \qquad \|Q|\Psi_\lambda\rangle\| = \sqrt{1-|\langle\Phi_0|\Psi_\lambda\rangle|^2} = \sin\theta(\lambda), \tag{12a}$$

$$\|P|\Phi_\lambda\rangle\| = |\langle\Phi_0|\Phi_\lambda\rangle| = \sqrt{C(\lambda)}, \qquad \|Q|\Phi_\lambda\rangle\| = \sqrt{1-|\langle\Phi_0|\Phi_\lambda\rangle|^2} = \sqrt{1-\mathcal{C}(\lambda)}, \tag{12b}$$

where the Bures angle $\theta(\lambda)$ and the ground state overlap $\mathcal{C}(\lambda)$ are introduced in Eqs. (6) and (4), respectively. The two vectors, $Q|\Psi_\lambda\rangle$ and $Q|\Phi_\lambda\rangle$, are not normalized; we defined the corresponding normalized vectors as

$$|\Phi_0^\perp(\lambda)\rangle := \frac{Q|\Psi_\lambda\rangle}{\|Q|\Psi_\lambda\rangle\|}, \qquad |\widetilde{\Phi}_0^\perp(\lambda)\rangle := \frac{Q|\Phi_\lambda\rangle}{\|Q|\Phi_\lambda\rangle\|}, \tag{13}$$

where the superscript $\perp$ indicates that these two normalized vectors are orthogonal to the initial state $|\Phi_0\rangle$. We introduce $\mathcal{D}(\lambda)$ to denote the overlap between the two normalized vectors, $|\Phi_0^\perp(\lambda)\rangle$ and $|\widetilde{\Phi}_0^\perp(\lambda)\rangle$,

$$\mathcal{D}(\lambda) := |\langle\Phi_0^\perp(\lambda)|\widetilde{\Phi}_0^\perp(\lambda)\rangle|^2, \tag{14}$$

and $\mathcal{D}_{\mathrm{un}}(\lambda)$ to denote the overlap between the two *unnormalized* vectors, $Q|\Psi_\lambda\rangle$ and $Q|\Phi_\lambda\rangle$,

$$\mathcal{D}_{\mathrm{un}}(\lambda) := \left|\langle\Psi_\lambda|Q|\Phi_\lambda\rangle\right|^2. \tag{15}$$

Note that the two overlaps, $\mathcal{D}(\lambda)$ and $\mathcal{D}_{\mathrm{un}}(\lambda)$, are not independent. They are related through

$$\sqrt{\mathcal{D}_{\mathrm{un}}(\lambda)} = \sin\theta(\lambda)\sqrt{1-\mathcal{C}(\lambda)}\sqrt{\mathcal{D}(\lambda)}. \tag{16}$$

Both the normalized overlap $\mathcal{D}(\lambda)$ and the unnormalized overlap $\mathcal{D}_{\mathrm{un}}(\lambda)$ play important roles in the following discussion.

It was found in Ref. [38] (for an alternative derivation using the projection–operator formalism, see Appendix A) that the difference between the adiabatic fidelity $\mathcal{F}(\lambda)$ (3) and the ground state overlap $\mathcal{C}(\lambda)$ (4) obeys the following inequality

$$|\mathcal{F}(\lambda)-\mathcal{C}(\lambda)| \quad\leq\quad \left|-\sin^2\theta(\lambda)\mathcal{C}(\lambda)+\mathcal{D}_{\mathrm{un}}(\lambda)\right|+2\cos\theta(\lambda)\sqrt{\mathcal{C}(\lambda)}\sqrt{\mathcal{D}_{\mathrm{un}}(\lambda)}, \tag{17}$$

where $\mathcal{D}_{\mathrm{un}}(\lambda)$ is defined in Eq. (15). To make further progress, the strategy made in Ref. [38] was to replace the normalized overlap $\mathcal{D}(\lambda)$ of Eq. (16) by its trivial upper bound 1, i.e., $\mathcal{D}(\lambda) \leq 1$, which renders the unnormalized overlap $\mathcal{D}_{\mathrm{un}}(\lambda)$ (16) bounded from above as follows

$$\sqrt{\mathcal{D}_{\mathrm{un}}(\lambda)} \quad\leq\quad \sin\theta(\lambda)\sqrt{1-\mathcal{C}(\lambda)}. \tag{18}$$

The rationale for adopting the trivial upper bound, $\mathcal{D}(\lambda) \leq 1$, is that, since presumably we have no knowledge about the overlap between the two normalized vectors, $|\Phi_0^\perp(\lambda)\rangle$ and $|\widetilde{\Phi}_0^\perp(\lambda)\rangle$ (13), we may simply replace their overlap with the trivial upper bound of their overlap. Applying the upper bound (18) to the inequality (17) yields

$$|\mathcal{F}(\lambda)-\mathcal{C}(\lambda)| \quad\leq\quad \sin^2\theta\,|1-2\mathcal{C}|+\sin(2\theta)\sqrt{\mathcal{C}}\sqrt{1-\mathcal{C}}. \tag{19}$$

Although the inequality (19) offers an improvement over the inequality (7) found in Ref. [37], it remains unclear why the values of the adiabatic fidelity $\mathcal{F}(\lambda)$ and the ground state overlap $\mathcal{C}(\lambda)$ are nearly identical when (i) the system size $N$ is sufficiently large (e.g., $N \geq 100$), or (ii) the evolution parameter $\lambda$ is small for any system size. In the present work, we address this question using the orthogonal decomposition formalism detailed in this section.

## 3 A motivating limit and interpretations

By inspecting Eq. (17), one observes that the least controlled piece in the inequality is the unnormalized overlap $\mathcal{D}_{\mathrm{un}}(\lambda)$ (15), which contains two factors [see Eq. (16)], namely, $\sqrt{D(\lambda)}$ and $\sin\theta(\lambda)\sqrt{1-\mathcal{C}(\lambda)}$. Among them, the trivial upper bound of $\mathcal{D}(\lambda)$ is employed in Ref. [38] to obtain universal upper bounds on $|\mathcal{F}(\lambda)-\mathcal{C}(\lambda)|$ [see Eq. (19)]. Therefore, the reason why the inequality (19) is insufficient to explain the smallness of $|\mathcal{F}(\lambda)-\mathcal{C}(\lambda)|$ may stem from the use of the trivial upper bound, $\mathcal{D}(\lambda) \leq 1$. To justify this claim, we simply look at the extreme limit:

$$\mathcal{D}_{\mathrm{un}}(\lambda) \to 0. \tag{20}$$

We will further elaborate on the orthogonality limit (20) later. For now, let us examine its consequences. Imposing the orthogonality limit (20) to the defining equations (11), the calculation of $|\mathcal{F}(\lambda)-\mathcal{C}(\lambda)|$ is fairly simple. First, we find from Eqs. (11) and (12) that

$$\mathcal{F}(\lambda) \to \cos^2\theta(\lambda)\mathcal{C}(\lambda). \tag{21}$$

It then follows that

$$|\mathcal{F}(\lambda) - \mathcal{C}(\lambda)| \to \sin^2\theta(\lambda)\mathcal{C}(\lambda). \tag{22}$$

Comparing the right side of Eq. (22) with that of (19) indicates that only a portion of Eq. (19) is retained on the right side of Eq. (22), resulting in a stronger upper bound on $|\mathcal{F}(\lambda) - \mathcal{C}(\lambda)|$.

The orthogonality limit (20), in view of Eq. (16), can be achieved by either

$$\text{(i) } \sin\theta(\lambda)\sqrt{1 - \mathcal{C}(\lambda)} \to 0, \qquad \text{or} \qquad \text{(ii) } \sqrt{D(\lambda)} \to 0. \tag{23}$$

Case (i) is satisfied if $\lambda$ is small. This is anticipated since if $\lambda$ is small, one expects that the Bures angle $\theta(\lambda)$ is still small as well, while both $\mathcal{C}(\lambda)$ and $\mathcal{D}(\lambda)$ remain close to one. To support this argument, we will present in Sec. 3.1 an explicit calculation based on the perturbative expansion in $\lambda$. As for case (ii), we shall see in Sec. 3.2 that it can be understood as a manifestation of almost-orthogonality occurring in the complementary subspace of the initial state $|\Phi_0\rangle$. We now elaborate on the two cases of Eq. (23) in turn.

## 3.1 Insights from perturbative expansion in $\lambda$

Here, we provide a further explanation for case (i) of Eq. (23). Our objective is to solve the instantaneous eigenvalue equation (2) and the time-dependent Schrödinger equation (1) perturbatively in $\lambda$ for the Hamiltonian $H_\lambda$ presented in Eq. (9), given the eigenvalue equation of $H_0$, $H_0|\chi_n\rangle = \varepsilon_n|\chi_n\rangle$, where $\{|\chi_n\rangle\}$ is a complete set of orthonormal eigenstates of $H_0$ with $|\chi_0\rangle \equiv |\Psi_0\rangle$ being its ground state and $n \in \{0, 1, \dots\}$ labels different eigenstates. One finds (refer to Appendix B for details) that, up to order $\lambda^2$, the adiabatic fidelity $\mathcal{F}(\lambda)$ (3) and the ground state overlap $\mathcal{C}(\lambda)$ (4) are identical and are independent of the driving rate $\Gamma$,

$$\mathcal{F}(\lambda) \simeq \mathcal{C}(\lambda) = 1 - \lambda^2 \sum_{n\neq 0} \frac{|V_{n0}|^2}{(\varepsilon_0 - \varepsilon_n)^2} + \mathcal{O}(\lambda^3), \tag{24}$$

where the matrix element $V_{nm} := \langle\chi_n|V|\chi_m\rangle$. The difference between $\mathcal{F}(\lambda)$ and $\mathcal{C}(\lambda)$ appears at order $\lambda^3$, $\mathcal{F}(\lambda) - \mathcal{C}(\lambda) = -\lambda^3 V_{00}\varepsilon_0/\Gamma^2 + \dots$ Leading order contributions for various quantities can also be obtained,

$$\sin\theta(\lambda) = \frac{\lambda^2}{2\Gamma}\Big(\sum_{n\neq 0}|V_{n0}|^2\Big)^{1/2} + \mathcal{O}(\lambda^3), \tag{25a}$$

$$\sqrt{\mathcal{D}_{\text{un}}(\lambda)} = \frac{\lambda^3}{2\Gamma}\sum_{n\neq 0}\frac{|V_{n0}|^2}{\varepsilon_n - \varepsilon_0} + \mathcal{O}(\lambda^4), \tag{25b}$$

$$\sin\theta(\lambda)\sqrt{1 - \mathcal{C}(\lambda)} = \mathcal{O}(\lambda^3), \tag{25c}$$

$$\sqrt{\mathcal{D}(\lambda)} = \mathcal{O}(1). \tag{25d}$$

We see that, for small $\lambda$, $\sqrt{\mathcal{D}_{\text{un}}(\lambda)}$ (16) is of order $\lambda^3$, which is attributed to the same order of small $\sin\theta(\lambda)\sqrt{1 - \mathcal{C}(\lambda)}$ (25c) since $\sqrt{\mathcal{D}(\lambda)}$ (25d) is of order one. However, as $\lambda$ continues to increase, the result from perturbation theory is insufficient to explain the smallness of $\sqrt{\mathcal{D}_{\text{un}}(\lambda)}$. Instead, when $\lambda$ is not small, the almost-orthogonality exhibited in the normalized overlap $\sqrt{\mathcal{D}(\lambda)}$, as shown in case (ii) of Eq. (23), should be taken into account.

## 3.2 Insights from almost-orthogonality in the complementary subspace under large system size

Case (ii) of Eq. (23) may be understood as follows. Let $\{|\Phi_0\rangle, |u_1\rangle, |u_2\rangle, \dots, |u_{n-1}\rangle\}$ be a complete set of $\lambda$-independent orthonormal basis in an $\mathfrak{n}$-dimensional Hilbert space $\mathscr{H}_{\mathfrak{n}}$. Since

both the time-evolved state $|\Psi_\lambda\rangle$ and the instantaneous ground state $|\Phi_\lambda\rangle$ are vectors in the full Hilbert space $\mathscr{H}_\mathfrak{n}$, it follows from the orthogonal decomposition (11) that the two normalized vectors, $|\Phi_0^\perp(\lambda)\rangle$ and $|\widetilde{\Phi}_0^\perp(\lambda)\rangle$ (13), are vectors lying in the subspace $\mathscr{H}_{\mathfrak{n}-1}^\perp$, where the codimension-1 Hilbert space $\mathscr{H}_{\mathfrak{n}-1}^\perp$ is spanned by $\{|u_1\rangle, |u_2\rangle, \ldots, |u_{\mathfrak{n}-1}\rangle\}$. When $\mathfrak{n}$ is large, the two normalized vectors, $|\Phi_0^\perp(\lambda)\rangle$ and $|\widetilde{\Phi}_0^\perp(\lambda)\rangle$, may be thought of as two independent *random vectors* in the Hilbert space $\mathscr{H}_{\mathfrak{n}-1}^\perp$ even though the vector $|\Phi_0^\perp(\lambda)\rangle$ undergoes dynamical evolution while the other vector $|\widetilde{\Phi}_0^\perp(\lambda)\rangle$ experiences adiabatic transformation. As a result, one would expect their overlap, $\mathcal{D}(\lambda)$ (14), to decay sufficiently fast with increasing $\mathfrak{n}$. Following literature in mathematics [54], we refer to this kind of orthogonal property as *almost-orthogonality*.

## 4 Reverse triangle inequalities

The discussion presented in the above section [see Eq. (22)] indicates that $\mathcal{F}(\lambda)$ is identical to $\cos^2\theta(\lambda)\mathcal{C}(\lambda)$ under the exact orthogonality limit (20). This observation motivates us to pursue bounds on the difference between them, namely, $|\sqrt{\mathcal{F}(\lambda)} - \cos\theta(\lambda)\sqrt{\mathcal{C}(\lambda)}|$. A useful tool for the present work is the following lemma.

**Lemma.** *The three real-valued quantities, $\sqrt{\mathcal{F}(\lambda)}$ (3), $\cos\theta(\lambda)\sqrt{\mathcal{C}(\lambda)}$ (4), and $\sqrt{\mathcal{D}_{\mathrm{un}}(\lambda)}$ (15), obey a set of (reverse) triangle inequalities,*

$$|\sqrt{\mathcal{F}(\lambda)} - \cos\theta(\lambda)\sqrt{\mathcal{C}(\lambda)}| \leq \sqrt{\mathcal{D}_{\mathrm{un}}(\lambda)}, \tag{26a}$$

$$\left|\sqrt{\mathcal{D}_{\mathrm{un}}(\lambda)} - \cos\theta(\lambda)\sqrt{\mathcal{C}(\lambda)}\right| \leq \sqrt{\mathcal{F}(\lambda)}, \tag{26b}$$

$$\left|\sqrt{\mathcal{D}_{\mathrm{un}}(\lambda)} - \sqrt{\mathcal{F}(\lambda)}\right| \leq \cos\theta(\lambda)\sqrt{\mathcal{C}(\lambda)}. \tag{26c}$$

Thus, $\sqrt{\mathcal{F}(\lambda)}$, $\cos\theta(\lambda)\sqrt{\mathcal{C}(\lambda)}$, and $\sqrt{\mathcal{D}_{\mathrm{un}}(\lambda)}$, form a triangle on a plane for all values of parameters. See Fig. 1 for an illustration.

*Proof.* First, we begin by considering the right side of Eq. (26a) with the help of Eq. (15),

$$\begin{aligned}
\sqrt{\mathcal{D}_{\mathrm{un}}(\lambda)} &= \left|\langle\Psi_\lambda|\Phi_\lambda\rangle - \langle\Psi_\lambda|\Phi_0\rangle\langle\Phi_0|\Phi_\lambda\rangle\right| \\
&\geq \left|\,|\langle\Psi_\lambda|\Phi_\lambda\rangle| - |\langle\Psi_\lambda|\Phi_0\rangle||\langle\Phi_0|\Phi_\lambda\rangle|\,\right| = \left|\sqrt{\mathcal{F}(\lambda)} - \cos\theta(\lambda)\sqrt{\mathcal{C}(\lambda)}\right|,
\end{aligned} \tag{27}$$

where we have used the reverse triangle inequality $|z - w| \geq ||z| - |w||$ for $z, w \in \mathbb{C}$. The inequality (26a) is thus established. Note that the right side of Eq. (27) may be interpreted as a lower bound on $\sqrt{\mathcal{D}_{\mathrm{un}}(\lambda)}$.

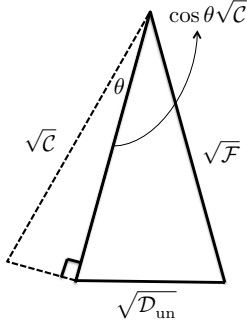

Figure 1: Depict of the triangle relationship (26) between the three real-valued quantities: $\sqrt{\mathcal{F}(\lambda)}$ (3), $\cos\theta(\lambda)\sqrt{\mathcal{C}(\lambda)}$ (4), and $\sqrt{\mathcal{D}_{\mathrm{un}}(\lambda)}$ (15), where $\theta(\lambda)$ is the Bures angle (6).

Second, we observe that there is an upper bound on $\sqrt{\mathcal{D}_{\text{un}}(\lambda)}$,

$$
\begin{aligned}
\sqrt{\mathcal{D}_{\text{un}}(\lambda)} &= \left| \langle \Psi_\lambda | \Phi_\lambda \rangle - \langle \Psi_\lambda | \Phi_0 \rangle \langle \Phi_0 | \Phi_\lambda \rangle \right| \\
&\leq |\langle \Psi_\lambda | \Phi_\lambda \rangle| + |\langle \Psi_\lambda | \Phi_0 \rangle||\langle \Phi_0 | \Phi_\lambda \rangle| = \sqrt{\mathcal{F}(\lambda)} + \cos\theta(\lambda)\sqrt{\mathcal{C}(\lambda)},
\end{aligned}
\tag{28}
$$

which is a consequence of the triangle inequality, $|z + w| \leq |z| + |w|$ for $z, w \in \mathbb{C}$.

Finally, combing the upper bound (28) with the lower bound (27) yields two two-sided bounds on $\sqrt{\mathcal{D}_{\text{un}}(\lambda)}$ (neglecting $\lambda$ to simplify notation),

$$
-\sqrt{\mathcal{F}} + \cos\theta\sqrt{\mathcal{C}} \quad \leq \quad \sqrt{\mathcal{D}_{\text{un}}} \quad \leq \quad \sqrt{\mathcal{F}} + \cos\theta\sqrt{\mathcal{C}}, \tag{29a}
$$
$$
\sqrt{\mathcal{F}} - \cos\theta\sqrt{\mathcal{C}} \quad \leq \quad \sqrt{\mathcal{D}_{\text{un}}} \quad \leq \quad \sqrt{\mathcal{F}} + \cos\theta\sqrt{\mathcal{C}}. \tag{29b}
$$

This completes the proof of Eqs. (26b) and (26c). □

The first triangle inequality (26a) provides a quantitative way to understand the closeness between $\mathcal{F}(\lambda)$ and $\mathcal{C}(\lambda)$ since

$$
\sqrt{\mathcal{F}} - \sqrt{\mathcal{C}} \quad \leq \quad \sqrt{\mathcal{F}} - \cos\theta(\lambda)\sqrt{\mathcal{C}} \quad \leq \quad \sqrt{\mathcal{D}_{\text{un}}}. \tag{30}
$$

Given that the unnormalized overlap $\sqrt{\mathcal{D}_{\text{un}}(\lambda)}$ acts as an upper bound, a small value for it, which can be achieved by the two cases of Eq. (23), suggests that the numerical difference between the adiabatic fidelity $\mathcal{F}(\lambda)$ and the ground state overlap $\mathcal{C}(\lambda)$ must be even smaller.

## 5 Illustrative example I: Non-interacting Hamiltonians

To illustrate our general analytical findings from Secs. 3 and 4, the remaining task is to explicitly express $\sqrt{\mathcal{D}_{\text{un}}(\lambda)}$, as defined in Eq. (15), in terms of the Bures angle $\theta(\lambda)$ and the ground state overlap $\mathcal{C}(\lambda)$ for specific models. This can be done analytically for non-interacting Hamiltonians for which one obtains [see Appendix C],

$$
\sqrt{\mathcal{D}_{\text{un}}(\lambda)} \simeq \cos\theta(\lambda)\sqrt{\mathcal{C}(\lambda)}\left|\sum_k A_k\right|, \qquad \text{where} \quad A_k := \frac{\langle \psi_\lambda(k)|(\mathbb{I}_k - p_k)|\phi_\lambda(k)\rangle}{\langle \psi_\lambda(k)|p_k|\phi_\lambda(k)\rangle}. \tag{31}
$$

Here, $|\psi_\lambda(k)\rangle$, $|\phi_\lambda(k)\rangle$, and $p_k$ are the single-body counterparts of $|\Psi_\lambda\rangle$, $|\Phi_\lambda\rangle$, and $P$ introduced in Sec. 2, respectively.

For concreteness, let us consider a time-dependent Rice-Mele model describing a system of fermions on a half-filled one-dimensional bipartite lattice with the Hamiltonian [37,55,56]

$$
H_{\text{RM}} := \sum_{j=1}^N \left[ -(J + U)a_j^\dagger b_j - (J - U)a_j^\dagger b_{j+1} + \text{h.c.} \right] + \sum_{j=1}^N \mu(\lambda)\left( a_j^\dagger a_j - b_j^\dagger b_j \right), \tag{32}
$$

where $N$, the number of lattice sites, is assumed to be even. Here, $a_j$ and $b_j$ are the fermionic annihilation operators on the $a$ and $b$ sublattices, respectively. For this model with $\mu(\lambda) = \lambda$, the ground state overlap $\mathcal{C}(\lambda)$ (4) and the function $\mathcal{R}(\lambda)$ (10) take the form shown in Eqs. (5) and (10) with $C_N = (16JU)^{-1}N$ and $\delta V_N = \sqrt{N}$.

We shall specialize to the case where $J = U = \text{constant}$ for which the summation in Eq. (31) can be evaluated in closed form [see Appendix C],

$$
\sqrt{\mathcal{D}_{\text{un}}(\lambda)} \simeq \sqrt{\mathcal{C}(\lambda)}\cos\theta(\lambda)\sin\theta(\lambda)\alpha(\lambda), \tag{33a}
$$
$$
\sqrt{\mathcal{D}(\lambda)} \simeq \sqrt{\mathcal{C}(\lambda)}\cos\theta(\lambda)\alpha(\lambda)/\sqrt{1 - \mathcal{C}(\lambda)}, \tag{33b}
$$

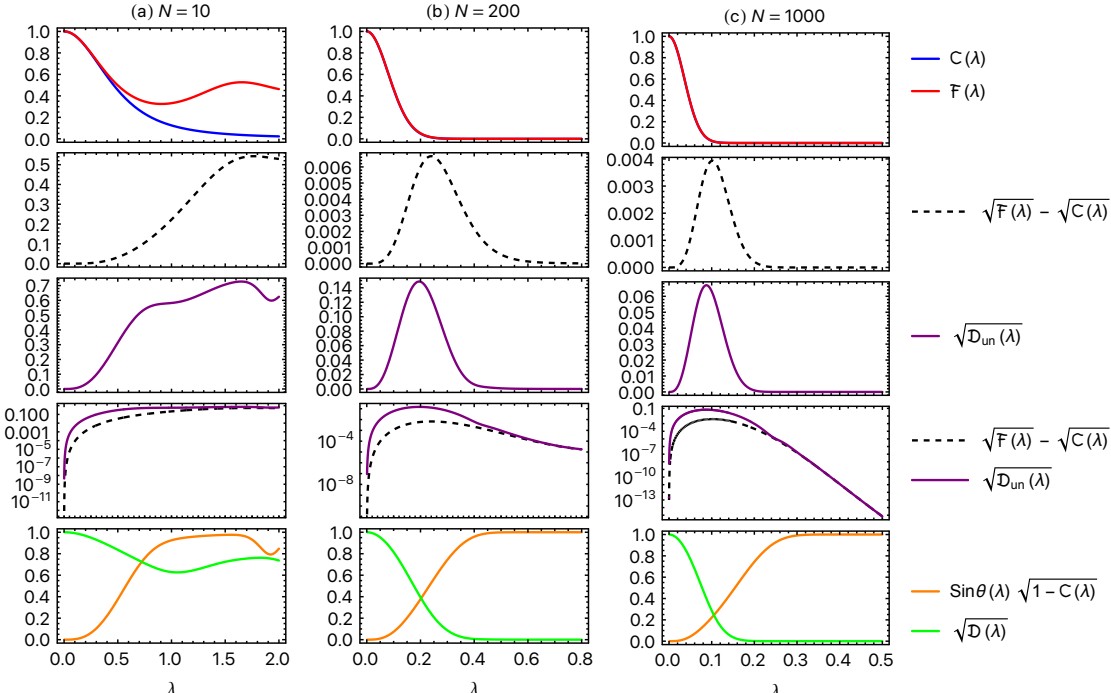

Figure 2: (Color online) Various quantities are calculated numerically for the driven Rice-Mele model (32) with the value of parameters shown in Eq. (35) for system size $N = 10, 200$ and 1000. The fourth row overlays the curves from the second and third rows, with the vertical axis on a logarithmic scale. Further explanation is provided in the main text.

where we have introduced an auxiliary function $\alpha(\lambda)$ for later convenience

$$\alpha(\lambda) := \sqrt{N} \sqrt{1 - \mathcal{C}(\lambda)^{1/N}}. \tag{33c}$$

We shall examine whether the explicit form of $\sqrt{\mathcal{D}_{\text{un}}(\lambda)}$ (33a) has the desired characteristics under the limit of small $\lambda$ or large $N$ as claimed previously in Sec. 3. First, it is readily checked that, for small $\lambda$, the leading order contribution to $\sqrt{\mathcal{D}_{\text{un}}(\lambda)}$ (33a) and $\sqrt{\mathcal{D}(\lambda)}$ (33b) are $\mathcal{O}(\lambda^3)$ and $\mathcal{O}(1)$, respectively, which is consistent with the results obtained from general perturbation theory as presented in Eq. (25). Second, note that the auxiliary function $\alpha(\lambda)$ (33c) scales at most with a rate of $\sqrt{N}$:

$$\alpha(\lambda) \leq \sqrt{cN\lambda^2} = \sqrt{-\ln \mathcal{C}(\lambda)}, \tag{34}$$

as a result of the inequality $1 - e^{-x} \leq x$ for all $x \in \mathbb{R}$. Therefore, we deduce from Eq. (33) that $\sqrt{\mathcal{D}_{\text{un}}(\lambda)} \sim \sqrt{\mathcal{D}(\lambda)} \sim \sqrt{\mathcal{C}(\lambda)}\sqrt{N} \to 0$ as $N \to \infty$, which is in agreement with the case (ii) of Eq. (23).

To numerically demonstrate our findings, we choose the following value of parameters

$$(J, U, \Gamma) = (0.4, 0.4, 0.7), \tag{35}$$

in the Hamiltonian $H_{\text{RM}}$ (32) as a representative example. In Fig. 2, we plot various quantities for the driven Rice-Mele model (32) with system size $N = 10, 200$, and 1000. In the first row, the adiabatic fidelity $\mathcal{F}(\lambda)$ and the ground state overlap $\mathcal{C}(\lambda)$ are indistinguishable for $N = 200$ and $N = 1000$. The second row shows that, for both $N = 200$ and $N = 1000$, the difference between $\sqrt{\mathcal{F}(\lambda)}$ and $\sqrt{\mathcal{C}(\lambda)}$ raises as $\lambda$ increases and then diminishes as $\lambda$ further increases. This bell-shaped curve of $\sqrt{\mathcal{F}(\lambda)} - \sqrt{\mathcal{C}(\lambda)}$ is in phase with the curve of the

unnormalized overlap $\sqrt{\mathcal{D}_{\mathrm{un}}(\lambda)}$ [third row]. This is consistent with Eq. (30), which states that $\sqrt{\mathcal{F}(\lambda)} - \sqrt{\mathcal{C}(\lambda)} \leq \sqrt{\mathcal{D}_{\mathrm{un}}(\lambda)}$ (see also the fourth row). In the fourth row, we overlay $\sqrt{\mathcal{F}(\lambda)} - \sqrt{\mathcal{C}(\lambda)}$ (dashed) and $\sqrt{\mathcal{D}_{\mathrm{un}}(\lambda)}$ (purple) in a single panel with the vertical axis on a logarithmic scale, making their separation visible. Since $\sqrt{\mathcal{D}_{\mathrm{un}}(\lambda)}$ can be factorized into two pieces, c.f. Eq. (16), the smallness of the monotonically *increasing* part of $\sqrt{\mathcal{F}(\lambda)} - \sqrt{\mathcal{C}(\lambda)}$ is attributed to the smallness of $\sin\theta(\lambda)\sqrt{1-\mathcal{C}(\lambda)}$ [fifth row]. Likewise, the monotonically *decreasing* part of $\sqrt{\mathcal{F}(\lambda)} - \sqrt{\mathcal{C}(\lambda)}$ is particularly small due to the almost-orthogonality occurring in the complementary space when $N$ is large, which is manifested by a small $\sqrt{\mathcal{D}(\lambda)}$ [fifth row]. By contrast, for $N = 10$ [see Fig. 2(a)], $\sqrt{\mathcal{F}(\lambda)} - \sqrt{\mathcal{C}(\lambda)}$ [second row of panel (a)] is monotonically increasing in most of the values of $\lambda$ and is small only in the region of small $\lambda$ (say, $\lambda \leq 0.2$). Again, this smallness of $\sqrt{\mathcal{F}(\lambda)} - \sqrt{\mathcal{C}(\lambda)}$ is related to the smallness of $\sin\theta(\lambda)\sqrt{1-\mathcal{C}(\lambda)}$ [fifth row of panel (a)]. When $\lambda$ further increases, however, the difference between $\sqrt{\mathcal{F}(\lambda)}$ and $\sqrt{\mathcal{C}(\lambda)}$ is notable since the normalized overlap $\sqrt{\mathcal{D}(\lambda)}$ [fifth row of panel (a)] does not exhibit almost-orthogonality for $N = 10$.

To further investigate the behavior of the normalized overlap $\sqrt{\mathcal{D}(\lambda)}$, we compare it with the ground state overlap $\sqrt{\mathcal{C}(\lambda)}$ in Fig. 3. Notably, both $\sqrt{\mathcal{D}(\lambda)}$ [green curve] and $\sqrt{\mathcal{C}(\lambda)}$ [blue curve] decay monotonically as $N$ and $\lambda$ increase. Moreover, $\sqrt{\mathcal{D}(\lambda)}$ exhibits a slower decay compared to $\sqrt{\mathcal{C}(\lambda)}$. For further comparison, the unnormalized overlap $\sqrt{\mathcal{D}_{\mathrm{un}}(\lambda)}$ [purple curve] is also depicted in Fig. 3.

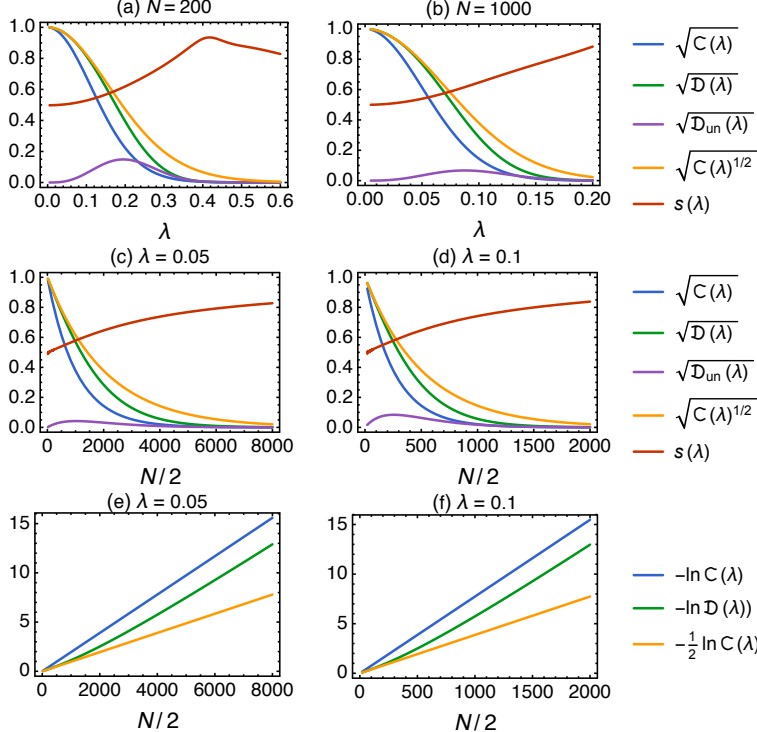

Figure 3: (Color online) Various quantities, $\sqrt{\mathcal{C}(\lambda)}$ (4), $\sqrt{\mathcal{D}(\lambda)}$ (14), $\sqrt{\mathcal{D}_{\mathrm{un}}(\lambda)}$ (15), $\sqrt{\mathcal{C}(\lambda)^{1/2}}$, and $s(\lambda)$ (39) for the driven Rice-Mele model (32) are plotted as a function of $\lambda$ or $N$.

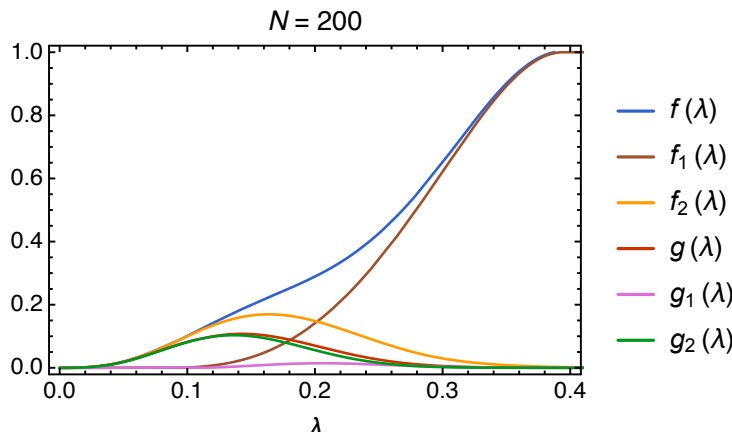

Figure 4: (Color online) Compare the behavior of the function $g(\lambda) = g_1(\lambda) + g_2(\lambda)$ (36) with the function $f(\lambda) = f_1(\lambda) + f_2(\lambda)$ (38) for $N = 200$.

Given the explicit form of $\sqrt{\mathcal{D}_{\mathrm{un}}(\lambda)}$ (33a), we may substitute it into Eq. (17) and apply the inequality of quantum speed limit (8) to obtain the following bound on $|\mathcal{F}(\lambda) - \mathcal{C}(\lambda)|$

$$|\mathcal{F}(\lambda) - \mathcal{C}(\lambda)| \leq g(\lambda), \qquad g(\lambda) := g_1(\lambda) + g_2(\lambda), \tag{36a}$$

$$g_1(\lambda) := \sin^2 \widetilde{\mathcal{R}}(\lambda) \mathcal{C}(\lambda) \left| -1 + \alpha(\lambda)^2 \right|, \tag{36b}$$

$$g_2(\lambda) := \sin\left( 2\widetilde{\widetilde{\mathcal{R}}}(\lambda) \right) \mathcal{C}(\lambda) \alpha(\lambda), \tag{36c}$$

where $\widetilde{\mathcal{R}}(\lambda)$ is defined in Eq. (8) and $\widetilde{\widetilde{\mathcal{R}}}(\lambda)$ is defined as

$$\widetilde{\widetilde{\mathcal{R}}}(\lambda) := \min\left( \mathcal{R}(\lambda), \frac{\pi}{4} \right). \tag{37}$$

Note that combing the inequality (36) with the defining range of $\mathcal{F}(\lambda)$, i.e., $\mathcal{F}(\lambda) \in [0, 1]$, yields the following two-sided bound on the adiabatic fidelity $\mathcal{F}(\lambda)$

$$\max(\mathcal{C}(\lambda) - g(\lambda), 0) \leq \mathcal{F}(\lambda) \leq \min(\mathcal{C}(\lambda) + g(\lambda), 1),$$

which provides a way to estimate the adiabatic fidelity $\mathcal{F}(\lambda)$ in terms of the ground state overlap $\mathcal{C}(\lambda)$ (4) and the function $\mathcal{R}(\lambda)$ (8b).

For comparison, let us revisit the inequality given by Eq. (19), which is derived from Eq. (17) by substituting the overlap $\sqrt{\mathcal{D}_{\mathrm{un}}(\lambda)}$ with its universal upper bound [see Eq. (18)]. When the quantum speed limit inequality from Eq. (8) is applied to bound the Bures angle $\theta(\lambda)$ in Eq. (19), the following inequality was derived in Ref. [38]:

$$|\mathcal{F}(\lambda) - \mathcal{C}(\lambda)| \leq f(\lambda), \qquad f(\lambda) := f_1(\lambda) + f_2(\lambda), \tag{38a}$$

$$f_1(\lambda) := \sin^2 \widetilde{\mathcal{R}}(\lambda) |1 - 2\mathcal{C}(\lambda)|, \tag{38b}$$

$$f_2(\lambda) := \sin(2\widetilde{\widetilde{\mathcal{R}}}(\lambda)) \sqrt{\mathcal{C}(\lambda)} \sqrt{1 - \mathcal{C}(\lambda)}, \tag{38c}$$

where $\widetilde{\mathcal{R}}(\lambda)$ and $\widetilde{\widetilde{\mathcal{R}}}(\lambda)$ are defined in Eq. (8) and Eq. (37), respectively.

In Fig. 4, we compare the two upper bounds on $|\mathcal{F}(\lambda) - \mathcal{C}(\lambda)|$: $g(\lambda) = g_1(\lambda) + g_2(\lambda)$ [Eq. (36)] and $f(\lambda) = f_1(\lambda) + f_2(\lambda)$ [Eq. (38)]. The former one corresponds to the case where the unnormalized overlap $\sqrt{\mathcal{D}_{\mathrm{un}}(\lambda)}$ takes the explicit form given in Eq. (33a), whereas the latter one is obtained by substituting $\sqrt{\mathcal{D}_{\mathrm{un}}(\lambda)}$ with its universal upper bound [Eq. (18)].

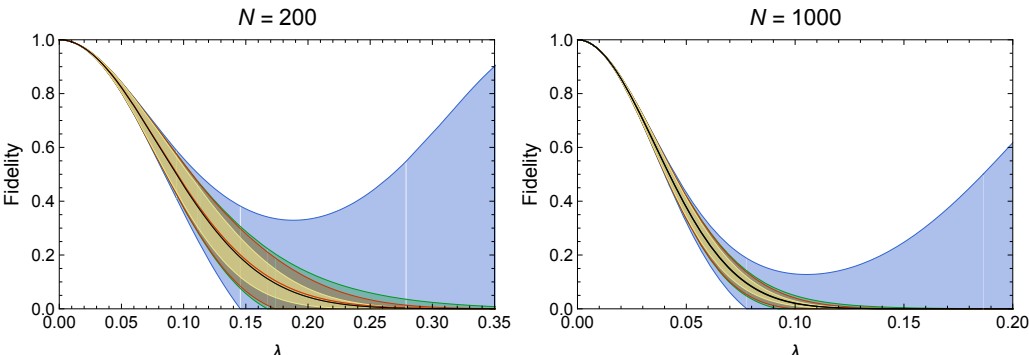

Figure 5: (Color online) Bounds on the adiabatic fidelity $\mathcal{F}(\lambda)$ for $N = 200$ and $N = 1000$ using Eq. (38) [blue-shaded region], Eq. (36) [red-shaded region], and Eq. (40) with $s = 1/2$ [green-shaded region] and $s = 1$ [yellow-shaded region]. For both figures, the actual adiabatic fidelity $\mathcal{F}(\lambda)$ is the black curve, while the red curve is for the ground state overlap $\mathcal{C}(\lambda)$, which is, however, not distinct from $\mathcal{F}(\lambda)$.

The result shows that the function $f(\lambda)$ [blue curve] increases monotonically with $\lambda$, whereas the function $g(\lambda)$ [red curve] exhibits a bell-shaped profile. This indicates that $g(\lambda)$ serves as a better upper bound compared to $f(\lambda)$. The enhanced performance of $g(\lambda)$ at larger $\lambda$ values stems from the function $g_1(\lambda)$ [Eq. (36b)], where the exponentially decaying factor $\mathcal{C}(\lambda)$ is extracted, offering a contrast to $f_1(\lambda)$ [Eq. (38b)]. Meanwhile, the difference between the function $g_2(\lambda)$ [green curve] and the function $f_2(\lambda)$ [orange curve] is not significant. Consequently, among the two inequalities, Eq. (36) and Eq. (38), the former provides a better estimate for the adiabatic fidelity $\mathcal{F}(\lambda)$.

In Fig. 5, we offer a comparison: estimates of the adiabatic fidelity $\mathcal{F}(\lambda)$ derived from Eq. (38) are represented by a blue-shaded region, while those from the improved inequality, Eq. (36), appear in a red-shaded region. The improvement in estimation achieved using the improved inequality is evident. Specifically, the improved estimate derived from Eq. (36) is effective even for a system size of $N = \mathcal{O}(10^2)$, which is the same large $N$ limit beyond which the ground state overlap $\mathcal{C}(\lambda)$ can be accurately approximated by a form of generalized orthogonality catastrophe [see Eq. (5)]. It is noteworthy that previous estimates on the adiabatic fidelity $\mathcal{F}(\lambda)$ obtained by Ref. [37] and Ref. [38] were only effectively applicable for larger system sizes, specifically $N = \mathcal{O}(10^4)$ and $N = \mathcal{O}(10^3)$, respectively.

## 6 Asymptotic form of the overlap $\mathcal{D}(\lambda)$ and implications

While the function $g(\lambda)$ [Eq. (36)] offers a better upper bound on $|\mathcal{F}(\lambda) - \mathcal{C}(\lambda)|$ than the function $f(\lambda)$ [Eq. (38)], determining the overlap $\mathcal{D}(\lambda)$ explicitly can be challenging for generic many-body systems. An explicit form of the overlap $\mathcal{D}(\lambda)$ is crucial for the enhancement in $g(\lambda)$. We thus seek for a universal scaling form of $\mathcal{D}(\lambda)$, upon which an estimate of upper bound on $|\mathcal{F}(\lambda) - \mathcal{C}(\lambda)|$ can be obtained by means of Eq. (17) without calculating $\mathcal{D}(\lambda)$ from scratch. In light of the reasoning of almost-orthogonality presented in Sec. 3.2, we consider the following ratio

$$s(\lambda) := \frac{\ln \mathcal{D}(\lambda)}{\ln \mathcal{C}(\lambda)}. \tag{39}$$

The value of $s(\lambda) \geq 0$ indicates how fast the overlap $\mathcal{D}(\lambda)$ decays compared to the overlap $\mathcal{C}(\lambda)$. If the ratio $s(\lambda)$ takes values in $[0, 1]$, then the overlap $\mathcal{D}(\lambda)$ decays not faster than the overlap $\mathcal{C}(\lambda)$ does.

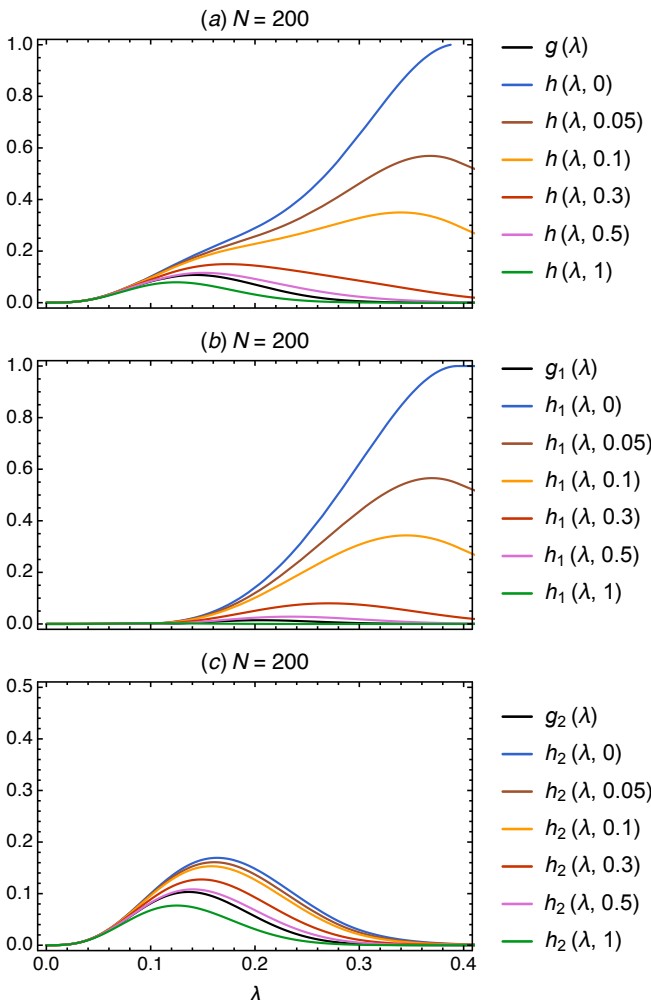

Figure 6: (Color online) Compare the behavior of the function $h(\lambda, s)$ (40a), $h_1(\lambda, s)$ (40b), and $h_2(\lambda, s)$ (40c) with different constant values of $s$. As for a comparison, the function $g(\lambda) = g_1(\lambda) + g_2(\lambda)$ (36) is also depicted.

By substituting $\mathcal{D}(\lambda)$ from Eq. (39), which is given by $\mathcal{D}(\lambda) = (\mathcal{C}(\lambda))^s$, into Eq. (17) and then applying the quantum speed limit inequality from (8), we obtain the following inequality:

$$|\mathcal{F}(\lambda) - \mathcal{C}(\lambda)| \quad \leq \quad h(\lambda, s) \equiv h_1(\lambda, s) + h_2(\lambda, s), \tag{40a}$$

$$h_1(\lambda, s) := \sin^2 \widetilde{\mathcal{R}}(\lambda)\mathcal{C}(\lambda)\left|1 - \mathcal{C}(\lambda)^{s-1} + \mathcal{C}(\lambda)^s\right|, \tag{40b}$$

$$h_2(\lambda, s) := \sin\left(2\widetilde{\widetilde{\mathcal{R}}}(\lambda)\right)\sqrt{\mathcal{C}(\lambda)^{s+1}}\sqrt{1 - \mathcal{C}(\lambda)}, \tag{40c}$$

where $\widetilde{\mathcal{R}}(\lambda)$ and $\widetilde{\widetilde{\mathcal{R}}}(\lambda)$ are defined in Eqs. (8a) and (37), respectively. The inequality (40) should be compared with that of Eq. (22), Eq. (36), and Eq. (38). It is worth noting that when $s = 0$, the function $h(\lambda, s = 0)$ simplifies to the function $f(\lambda)$ in Eq. (38). Observe that the function $h(\lambda, s)$ from Eq. (40) is expressed as $\sin \mathcal{R}(\lambda)\mathcal{C}(\lambda) \times (\cdots)$, where the terms inside the parenthesis scale at most polynomially in $N$ and $\lambda$ provided $s$ is not too small. The dominant factor in the function $h(\lambda, s)$ for large $N$ and large $\lambda$ is the $\sin \mathcal{R}(\lambda)\mathcal{C}(\lambda)$ component. This dominance is also observed in Eq. (22) and in the function $g(\lambda)$ from Eq. (36). However, this is not the case for the function $f(\lambda)$ in Eq. (38). In essence, only the behavior of the function $h_1(\lambda, s)$ at large $\lambda$ values determines whether $h(\lambda, s)$ can serve as a suitable upper bound.

Although the value of $s$ in Eq. (40) typically depends on both $\lambda$ and $N$, it is possible to approximate it using specific constant values, thereby facilitating the use of Eq. (40) in the estimation of adiabatic fidelity $\mathcal{F}(\lambda)$. To illustrate this, we shall revisit the driven Rice-Mele model presented in Sec. 5. In Fig. 6, we plot $h(\lambda,s) = h_1(\lambda,s) + h_2(\lambda,s)$ (40) as a function of $\lambda$ for different constant values of $s = 0, 0.05, 0.1, 0.3, 0.5, 1$. We observe that as long as the value of $s$ is not too small, say, $s \gtrsim 0.3$, the function $h_1(\lambda,s)$ [see Fig. 6(b)] decays quickly at large $\lambda$, which improves the tail behavior of the function $h(\lambda,s)$ [see Fig. 6(a)]. On the other hand, the behavior of $h_2(\lambda,s)$ [see Fig. 6(c)] does not change significantly as the value of $s$ varies. Nevertheless, as long as the value of $s$ is sufficiently large, the function $h(\lambda,s)$ is dominated by $h_2(\lambda,s)$ and serves as a strong upper bound on $|\mathcal{F}(\lambda) - \mathcal{C}(\lambda)|$. In Fig. 5, we plot bounds on the adiabatic fidelity $\mathcal{F}(\lambda)$ for $N = 200$ and $N = 1000$ using Eq. (40) with $s = 1/2$ [green-shaded region] and $s = 1$ [yellow-shaded region]. Note that the case of $s = 0$ reduces to the inequality (38) [blue-shaded region]. One observes that the difference between the green-shaded area and the red-shaded area is not significant. That is to say, the result of taking $s = 1/2$, namely, taking $\mathcal{D}(\lambda) \approx \sqrt{\mathcal{C}(\lambda)}$, is very close to the result obtained from the inequality (36), which is derived using an explicit form of the normalized overlap $\mathcal{D}(\lambda)$ (33b). Consequently, we also plot $\sqrt{\mathcal{C}(\lambda)}$ and $s(\lambda)$ (39) in Fig. 3, which shows that $\sqrt{\mathcal{C}(\lambda)}$ is slightly larger than $\mathcal{D}(\lambda)$. Nevertheless, for the purpose of estimating the adiabatic fidelity $\mathcal{F}(\lambda)$ using the inequality (40), replacing $s(\lambda)$ by a constant value (such as $1/2$) may be a good approximation.

**Condition for adiabaticity breakdown revisited**

Before concluding this section, let us discuss implications of a condition for adiabaticity breakdown using inequality (30). Recall that a large class of driven many-body systems exists for which the condition

$$\delta V_N / C_N = 0, \quad \text{as} \quad N \to \infty, \tag{41}$$

holds [37]. Here, $\delta V_N$ is introduced in Eq. (10), and $C_N$ refers to the exponent given in Eq. (5). Define $\lambda_*$ as an *adiabatic mean free path* so that $\mathcal{F}(\lambda) \geq e^{-1}$ for $\lambda \leq \lambda_*$. Here, $\lambda_*$ is determined by $\mathcal{C}(\lambda_*) = 1/e$. The relation $\mathcal{R}(\lambda_*) = \delta V_N / (2\Gamma C_N)$ then follows from Eq. (10). It was shown in Ref. [37] that, in order to avoid adiabaticity breakdown, the driving rate $\Gamma$ of driven many-body systems must scale down with increasing system size $N$. Using inequality (30), we find that quantum adiabaticity is maintained if $\Gamma \leq \Gamma_N$ with [see Appendix D for a derivation]

$$\Gamma_N := \frac{1}{2} \frac{\delta V_N}{C_N} \frac{1}{\left(1 - \epsilon - e^{-1/2}\right)^2} M(s_*), \tag{42}$$

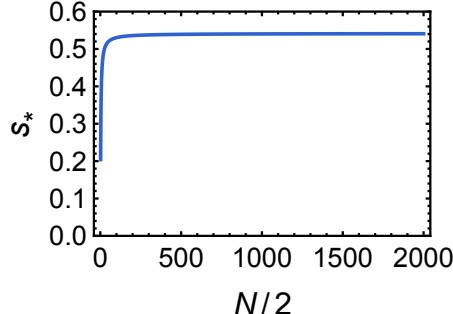

Figure 7: (Color online) The ratio $s_*$, defined in Eq. (39) with $\lambda = \lambda_*$ (so that $\mathcal{C}(\lambda_*) = 1/e$), is calculated numerically for the driven Rice-Mele model (32) with the value of parameters shown in Eq. (35). Asymptotically, $s_* = \mathcal{O}(1)$ as $N \to \infty$.

where $\epsilon \in [0,1]$, $M(s_*) := \sqrt{1-e^{-1}}\, e^{-s_*/2}$ and $s_* := s(\lambda_*) = -\ln \mathcal{D}(\lambda_*)$. Consequently, the asymptotic form of $M(s_*)$ as $N \to \infty$ contributes to a multiplicative modification to the scaling form $\Gamma_N \sim \delta V_N / C_N$ found previously in Refs. [37,38]. Two key observations can be made: (i) If $s_* = \mathcal{O}(1)$ as $N \to \infty$, then $M(s_*) = \mathcal{O}(1)$. (ii) For $s_* = \mathcal{O}(N^c)$ with $c$ being a real number as $N \to \infty$, then $M(s_*) = \mathcal{O}(e^{-N^c})$. Further, it is verified that when the leading asymptotics of $-\ln \mathcal{D}(\lambda)$ is proportional to that of $-\ln \mathcal{C}(\lambda)$ as $N \to \infty$, then the condition $s_* = \mathcal{O}(1)$ is satisfied. Specifically, for the driven Rice-Mele model (32), we determine that $s_* = \mathcal{O}(1)$ as $N \to \infty$ [refer to Fig. 7], which implies $M(s_*) = \mathcal{O}(1)$.

# 7 Illustrative example II: Interacting fermions

Up to this point, our general results from Secs. 3 and 4, have been illustrated using the driven Rice-Mele model (32), a model of quadratic fermions characterized by the unique property where the underlying Hilbert space is constructed as a direct product of single-particle states. A pivotal question emerges regarding whether the phenomenon of almost-orthogonality between vectors in the complement of the initial state exists in typical many-body systems, especially those governed by nonintegrable interacting Hamiltonians. In this section, we respond to the question affirmatively. To demonstrate this, we analyze an interacting Kitaev chain model defined by the following Hamiltonian

$$H_K := \sum_{j=1}^{N} \left[ \left( -J c_j^\dagger c_{j+1} + \Delta c_j^\dagger c_{j+1}^\dagger + \text{h.c.} \right) + V n_j n_{j+1} \right] + \sum_{j=1}^{N} \mu(\lambda) n_j, \tag{43}$$

where $n_j := c_j^\dagger c_j$ is the number operator of fermions at lattice site $j$, $N$ the number of lattice sites, $J$ the hopping amplitude, $\Delta$ the superconducting pairing amplitude, $V$ the strength of nearest-neighbor Coulomb repulsion, and $\mu(\lambda) = \mu_0 \lambda$ (with $\mu_0 \in \mathbb{R}$) the time-dependent

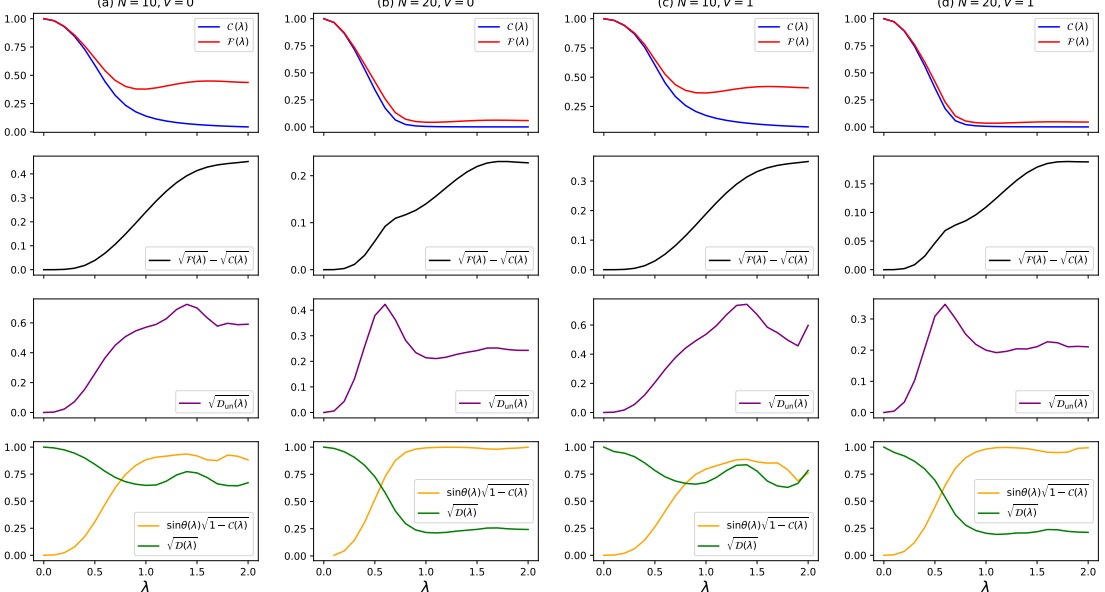

Figure 8: (Color online) Various quantities are calculated numerically for the driven interacting Kitaev chain model (43) with the value of parameters shown in Eq. (44) for system size $N = 10$ and $N = 20$ with interaction strength $V = 0$ and $V = 1$. Further explanation is provided in the main text.

chemical potential. If $V = 0$, the Hamiltonian (43) reduces to that of the Kitaev model of one-dimensional $p$-wave superconductors [57]. We shall consider the Hamiltonian (43) with periodic boundary conditions and the sector of odd fermion parity. To ensure that our numerical simulation results reflect generic features, we avoid selecting parameter values corresponding to solvable points [58]. For concreteness, we choose the following parameter values

$$(J, \Delta, \mu_0, \Gamma) = (1.0, 0.8, 3.0, 1.0). \tag{44}$$

In Fig. 8, we plot various quantities as functions of $\lambda$ for the driven interacting Kitaev model (43). Specifically, panels (a) and (c) depict results for a system size of $N = 10$, while panels (b) and (d) illustrate results for $N = 20$. The interaction strength $V = 0$ is chosen for panels (a) and (b), and $V = 1$ for panels (c) and (d). The behavior of curves in panel (a) is quantitatively similar to those of the driven Rice-Mele model with $N = 10$ presented in Fig. 2(a). For the increased system size $N = 20$, which approaches the computational limits of the exact diagonalization method, the notable characteristic is the expedited decay of $\mathcal{F}(\lambda)$, $\mathcal{C}(\lambda)$, and $\mathcal{D}(\lambda)$, as illustrated in Fig. 8(b). This trend aligns with observations for the driven Rice-Mele model shown in Fig. 2. Panels (c) and (d) of Fig. 8 represent the case of the driven *interacting* Kitaev model with $V = 1$ for system sizes $N = 10$ and $N = 20$, respectively. Compared with their non-interacting counterparts [namely, panels (a) and (b)], the differences between panels (c) and (a), as well as between panels (d) and (b), are not significant. Hence, we anticipate that the phenomenon of almost-orthogonality between vectors in the complement of the initial state should exist even in the presence of interacting Hamiltonians. To further elucidate this point, we plot in Fig. 9 the quantities $-\ln \mathcal{C}(\lambda)$ (4) and $-\ln \mathcal{D}(\lambda)$ (14) for the driven inter-

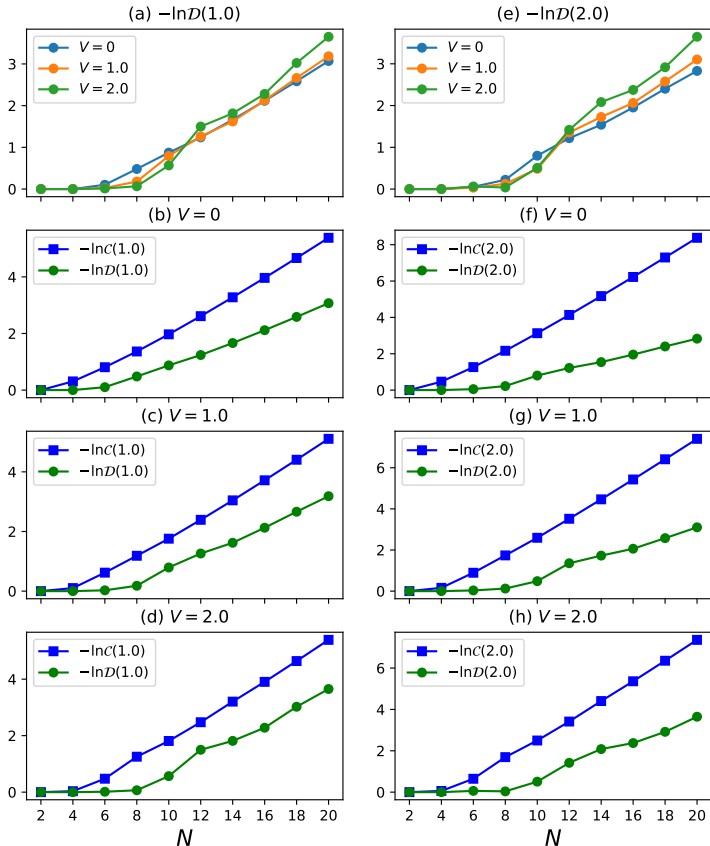

Figure 9: (Color online) The quantities $-\ln \mathcal{C}(\lambda)$ (4) and $-\ln \mathcal{D}(\lambda)$ (14) for the driven interacting Kitaev model (43) are plotted as a function of $N$ for different values of $\lambda$ ($\lambda = 1$ in the left column and $\lambda = 2$ in the right column) and $V = 0, 1, 2$.

acting Kitaev model (43) as a function of system size $N$ for $\lambda = 1, 2$ and interaction strength $V = 0, 1, 2$. According to all the panels of Fig. 9, the normalized overlap $\mathcal{D}(\lambda)$ decays with increasing system size $N$, a characteristic of almost-orthogonality in complementary subspace.

As a final note, we would like to highlight two additional notable numerical findings from Fig. 9. First, the ground state overlap $\mathcal{C}(\lambda)$ decays faster than the normalized overlap $\mathcal{D}(\lambda)$, a feature found similarly in the driven Rice-Mele model (see Fig. 3). Second, the decay exponent of $\mathcal{D}(\lambda)$ increases with increasing interaction strength $V$. This observation may suggest that the presence of interactions leads to a distribution of vectors in the underlying Hilbert space that appears more "random", resulting in a more rapid decay of the overlap $\mathcal{D}(\lambda)$ as the interaction strength increases (for a fixed system size). The extent to which these two observations are universal, however, warrants further investigation.

## 8 Summary and outlook

This study elucidates the reasons behind the frequent observation in quantum many-body systems where the adiabatic fidelity, $\mathcal{F}(\lambda)$, and the overlap between the initial and instantaneous ground states, $\mathcal{C}(\lambda)$, exhibit nearly identical values in numerous instances, especially in regions with small evolution parameters or large system sizes. While this observation can be rationalized in the region of small evolution parameter $\lambda$ using detailed perturbation theory, which shows that the difference between $\mathcal{F}(\lambda)$ and $\mathcal{C}(\lambda)$ only appears at the $\lambda^3$ order, a thorough explanation for the region of large system size fundamentally hinges on an intrinsic property of quantum many-body systems: *the almost-orthogonality of random vectors*. Specifically, this work details how the almost-orthogonality in the complementary space of the initial state, as exhibited by an exponentially decaying normalized overlap $\mathcal{D}(\lambda)$ (14) and a small value of unnormalized overlap $\mathcal{D}_{\text{un}}(\lambda)$ (15), controls an upper bound on the difference between $\sqrt{\mathcal{F}(\lambda)}$ and $\sqrt{\mathcal{C}(\lambda)}$ [see Eq. (30)]. To support these general, model-independent findings, numerical studies were conducted on both a driven Rice-Mele model and a driven interacting Kitaev model.

As a secondary result, our study provides improved estimates for the adiabatic fidelity $\mathcal{F}(\lambda)$. These estimates rely on an explicit representation of the normalized overlap $\mathcal{D}(\lambda)$ as a specific function of the ground state overlap $\mathcal{C}(\lambda)$, leveraging the concept of almost-orthogonality. Using the driven Rice-Mele model as an illustration, we demonstrated that these refined estimates perform well down to system sizes $N = \mathcal{O}(10^2)$—the same threshold beyond which the ground state overlap $\mathcal{C}(\lambda)$ is accurately captured by a generalized orthogonality–catastrophe form. These results distinctly outperform the earlier estimates of Refs. [37,38], which achieve good accuracy mainly for $N \gtrsim 10^3$, marking a notable advance in the precision and practical utility of adiabatic-fidelity estimates.

There are two key observations identified through our numerical analysis that warrant further attention: (i) The ground state overlap $\mathcal{C}(\lambda)$ decays more rapidly than the normalized overlap $\mathcal{D}(\lambda)$, and (ii) the decay exponent of $\mathcal{D}(\lambda)$ increases with increasing interaction strength. Future work should investigate whether these two observations persist across a broader range of models and explore the potential connection between the degree of *almost-orthogonality* and the dichotomy between chaotic and integrable many-body systems.

We conclude by noting that it would be worthwhile to pursue a rigorous proof (similar to that in Ref. [53]) to confirm the existence of subspace almost-orthogonality. Additionally, it would be intriguing to explore how subspace almost-orthogonality might affect other driven many-body systems known to exhibit orthogonality catastrophe in the full Hilbert space, such as those with time-dependent impurities [59] or undergoing quantum quenches [60].

## Acknowledgments

The numerical calculations of Sec. 7 were accomplished using QuSpin [61, 62].

**Funding information** This work is part of the Adiabatic Protocols in Extended Quantum Systems project, funded by the Dutch Research Council (NWO) under Project No 680-91-130. During the revision of this work, J.-H. C. was supported by the National Science and Technology Council (NSTC) of Taiwan under Grant No. 113-2112-M-008-037-MY3.

## A  Alternative derivation of Eq. (17)

We rederive inequality (17)—originally obtained in Ref. [38]—using a more compact projection-operator technique. First, we compute $\mathcal{F}(\lambda)$ using Eq. (11),

$$
\begin{aligned}
\mathcal{F}(\lambda) &= |\langle \Phi_\lambda |(P+Q)|\Psi_\lambda\rangle|^2 \\
&= |\langle \Phi_\lambda |P|\Psi_\lambda\rangle + \langle \Phi_\lambda |Q|\Psi_\lambda\rangle|^2 \\
&= |\langle \Phi_\lambda |P|\Psi_\lambda\rangle|^2 + |\langle \Phi_\lambda |Q|\Psi_\lambda\rangle|^2 + 2\Re\left(\langle \Psi_\lambda |P|\Phi_\lambda\rangle\langle \Phi_\lambda |Q|\Psi_\lambda\rangle\right).
\end{aligned}
\tag{A.1}
$$

Combing (i) Eq. (A.1), (ii) the triangle inequality for absolute value, (iii) the inequality $\Re(z) \le |z|$ for $z \in \mathbb{C}$, and (iv) Eqs. (12) and (14) yields the following chain of inequalities

$$
\begin{aligned}
|\mathcal{F}(\lambda) - \mathcal{C}(\lambda)| &\overset{(i)}{=} \left| |\langle \Phi_\lambda |P|\Psi_\lambda\rangle|^2 - \mathcal{C}(\lambda) + |\langle \Phi_\lambda |Q|\Psi_\lambda\rangle|^2 + 2\Re\left(\langle \Psi_\lambda |P|\Phi_\lambda\rangle\langle \Phi_\lambda |Q|\Psi_\lambda\rangle\right) \right| \\
&\overset{(ii)}{\le} \left| |\langle \Phi_\lambda |P|\Psi_\lambda\rangle|^2 - \mathcal{C}(\lambda) + |\langle \Phi_\lambda |Q|\Psi_\lambda\rangle|^2 \right| + 2\left| \Re\left(\langle \Psi_\lambda |P|\Phi_\lambda\rangle\langle \Phi_\lambda |Q|\Psi_\lambda\rangle\right) \right| \\
&\overset{(iii)}{\le} \left| |\langle \Phi_\lambda |P|\Psi_\lambda\rangle|^2 - \mathcal{C}(\lambda) + |\langle \Phi_\lambda |Q|\Psi_\lambda\rangle|^2 \right| + 2\left| \langle \Psi_\lambda |P|\Phi_\lambda\rangle\right|\left|\langle \Phi_\lambda |Q|\Psi_\lambda\rangle \right| \\
&\overset{(iv)}{=} \left| -\sin^2\theta(\lambda)\mathcal{C}(\lambda) + \mathcal{D}_{\mathrm{un}}(\lambda) \right| + 2\cos\theta(\lambda)\sqrt{\mathcal{C}(\lambda)}\sqrt{\mathcal{D}_{\mathrm{un}}(\lambda)}.
\end{aligned}
\tag{A.2}
$$

### A further alternative derivation

Alternatively, the inequality (17) can also be obtained using the set of triangle inequality (26). We begin with the upper bound on $\sqrt{\mathcal{F}(\lambda)}$ from Eq. (26a), $\sqrt{\mathcal{F}(\lambda)} \le \cos\theta(\lambda)\sqrt{\mathcal{C}(\lambda)} + \sqrt{\mathcal{D}_{\mathrm{un}}(\lambda)}$, and then calculate $\mathcal{F}(\lambda) - \mathcal{C}(\lambda)$,

$$
\begin{aligned}
\mathcal{F}(\lambda) - \mathcal{C}(\lambda) &\le -\sin^2\theta(\lambda)\mathcal{C}(\lambda) + \mathcal{D}_{\mathrm{un}}(\lambda) + 2\cos\theta(\lambda)\sqrt{\mathcal{C}(\lambda)}\sqrt{\mathcal{D}_{\mathrm{un}}(\lambda)} \\
&\le \left| -\sin^2\theta(\lambda)\mathcal{C}(\lambda) + \mathcal{D}_{\mathrm{un}}(\lambda) \right| + 2\cos\theta(\lambda)\sqrt{\mathcal{C}(\lambda)}\sqrt{\mathcal{D}_{\mathrm{un}}(\lambda)},
\end{aligned}
\tag{A.3}
$$

where the last step is obtained after using the inequality, $x + y \le |x + y|$ for $x, y \in \mathbb{R}$. Similarly, we consider the lower bound on $\sqrt{\mathcal{F}(\lambda)}$ from Eq. (26b), $\sqrt{\mathcal{F}(\lambda)} \ge \left| \sqrt{\mathcal{D}_{\mathrm{un}}(\lambda)} - \cos\theta(\lambda)\sqrt{\mathcal{C}(\lambda)} \right|$, and then calculate $\mathcal{F}(\lambda) - \mathcal{C}(\lambda)$,

$$
\begin{aligned}
\mathcal{F}(\lambda) - \mathcal{C}(\lambda) &\ge -\sin^2\theta(\lambda)\mathcal{C}(\lambda) + \mathcal{D}_{\mathrm{un}}(\lambda) - 2\cos\theta(\lambda)\sqrt{\mathcal{C}(\lambda)}\sqrt{\mathcal{D}_{\mathrm{un}}(\lambda)} \\
&\ge -\left| -\sin^2\theta(\lambda)\mathcal{C}(\lambda) + \mathcal{D}_{\mathrm{un}}(\lambda) \right| - 2\cos\theta(\lambda)\sqrt{\mathcal{C}(\lambda)}\sqrt{\mathcal{D}_{\mathrm{un}}(\lambda)},
\end{aligned}
\tag{A.4}
$$

where the last step is obtained after using the inequality, $x + y \ge -|x + y|$ for $x, y \in \mathbb{R}$. Combing Eqs. (A.3) with (A.4) yields the inequality (17).

# B  Perturbative expansion in $\lambda$

A detailed derivation of Eqs. (24) and (25) is provided.

## B.1  For instantaneous ground state

We want to solve the instantaneous eigenvalue equation (2) perturbatively in $\lambda$,

$$H_\lambda|\Phi_\lambda\rangle = E_{\text{GS},\lambda}|\Phi_\lambda\rangle\,, \qquad H_\lambda = H_0 + \lambda V\,, \tag{B.1}$$

given $H_0|\chi_n\rangle = \varepsilon_n|\chi_n\rangle$, where $|\Phi_\lambda\rangle$ is the ground state of $H_\lambda$, and $\{|\chi_n\rangle\}$ is a set of the complete orthonormal eigenstates of $H_0$ with $|\chi_0\rangle \equiv |\Psi_0\rangle$ being the ground state of $H_0$ and $n = 0, 1, \ldots$ labels distinct eigenstates. Apply the standard Rayleigh-Schrödinger perturbation theory up to order $\lambda^2$ yields the following series,

$$|\Phi_\lambda\rangle = \left(1 - \frac{\lambda^2}{2}\sum_{n\neq 0}\frac{|V_{n0}|^2}{(\varepsilon_0-\varepsilon_n)^2}\right)|\chi_0\rangle + \lambda\sum_{n\neq 0}\frac{V_{n0}}{\varepsilon_0-\varepsilon_n}|\chi_n\rangle$$
$$+ \lambda^2\sum_{n\neq 0}\frac{1}{\varepsilon_0-\varepsilon_n}\left(\sum_{m\neq 0}\frac{V_{nm}V_{m0}}{\varepsilon_0-\varepsilon_m} - \frac{V_{00}V_{n0}}{\varepsilon_0-\varepsilon_n}\right)|\chi_n\rangle + \ldots\,,$$

where $V_{nm} := \langle\chi_n|V|\chi_m\rangle$. Hence, the following inner products are obtained,

$$\langle\chi_0|\Phi_\lambda\rangle = 1 - \frac{\lambda^2}{2}\sum_{n\neq 0}\frac{|V_{n0}|^2}{(\varepsilon_0-\varepsilon_n)^2}$$
$$+ \lambda^3\left(\sum_{n\neq 0}\frac{V_{00}|V_{n0}|^2}{(\varepsilon_0-\varepsilon_n)^3} - \sum_{n\neq 0}\frac{1}{(\varepsilon_0-\varepsilon_n)^2}\sum_{m\neq 0}\frac{\Re(V_{nm}^*V_{m0}^*V_{n0})}{\varepsilon_0-\varepsilon_m}\right) + \ldots\,, \tag{B.2a}$$

$$\langle\chi_{n\neq 0}|\Phi_\lambda\rangle = \lambda\frac{V_{n0}}{\varepsilon_0-\varepsilon_n} + \lambda^2\frac{1}{\varepsilon_0-\varepsilon_n}\left(\sum_{m\neq 0}\frac{V_{nm}V_{m0}}{\varepsilon_0-\varepsilon_m} - \frac{V_{00}V_{n0}}{\varepsilon_0-\varepsilon_n}\right) + \ldots \tag{B.2b}$$

Notice that both inner products, $\langle\chi_0|\Phi_\lambda\rangle$ and $\langle\chi_{n\neq 0}|\Phi_\lambda\rangle$, are real-valued.

## B.2  For time-evolved state

We want to solve the time-dependent Schrödinger equation (1) perturbatively,

$$i\Gamma\partial_\lambda|\Psi_\lambda\rangle = \left(H_0 + \lambda V\right)|\Psi_\lambda\rangle\,, \qquad |\Psi_0\rangle = |\chi_0\rangle\,, \tag{B.3}$$

given $H_0|\chi_n\rangle = \varepsilon_n|\chi_n\rangle$. The following perturbative expansion in $\lambda$ (i.e., reduced time) is different from the usual time-dependent perturbation theory in which the expansion parameter is time-independent. Hence, we provide some details for our perturbative approach. Generically, we can decompose $|\Psi_\lambda\rangle$ as

$$|\Psi_\lambda\rangle = \sum_n C_n(\lambda)\exp\left(-i\lambda\varepsilon_n/\Gamma\right)|\chi_n\rangle\,, \tag{B.4}$$

with $\lambda$-dependent coefficients $C_n(\lambda)$ from which a factor $\exp\left(-i\lambda\varepsilon_n/\Gamma\right)$ has been extracted for later convenience. Since $|\Psi_0\rangle = |\chi_0\rangle$, we have $C_n(0) = \delta_{n0}$. Bring the decomposition (B.4) into Eq. (B.3) yields a first-order differential equation for $C_n(\lambda)$,

$$\partial_\lambda C_m(\lambda) = \sum_n C_n(\lambda)\exp\left(i\lambda\omega_{mn}/\Gamma\right)\lambda\frac{V_{mn}}{i\Gamma}\,, \tag{B.5}$$

where $\omega_{mn} := \varepsilon_m - \varepsilon_n$. Now, as we are interested in small $\lambda$ region, we may expand $C_n(\lambda)$ in power series of $\lambda$, namely, $C_n(\lambda) = \sum_{j=0}^{\infty} \lambda^j C_n^{(j)} = \delta_{n0} + \sum_{j=1}^{\infty} \lambda^j C_n^{(j)}$. We shall also expand the $\exp(i\lambda\omega_{mn}/\Gamma)$ factor in powers of $\lambda$. The differential equation (B.5) then reads

$$\sum_{j=1}^{\infty} j\lambda^{j-1} C_m^{(j)} = \sum_n \left( \sum_{j=0}^{\infty} \lambda^j C_n^{(j)} \right) \left( \sum_{\ell=0}^{\infty} \frac{1}{\ell!} \left(i\lambda\omega_{mn}/\Gamma\right)^\ell \right) \lambda \frac{V_{mn}}{i\Gamma}. \tag{B.6}$$

We now match terms for each order in $\lambda$. One finds that $C_m^{(1)} = 0$ and, generically, the term in the $k$-th order of $\lambda$ with $k \geq 1$ reads,

$$\lambda^k : \quad C_m^{(k+1)} = \sum_n \sum_{\ell=0}^{k-1} C_n^{(k-\ell-1)} \frac{1}{\ell!} \left(i\omega_{mn}/\Gamma\right)^\ell \frac{V_{mn}}{(k+1)i\Gamma}.$$

The first few leading order contributions are

$$C_m^{(2)} = \frac{V_{m0}}{2i\Gamma}, \qquad C_m^{(3)} = \frac{\omega_{m0} V_{m0}}{3\Gamma^2}, \qquad C_m^{(4)} = -\sum_n \frac{V_{n0} V_{mn}}{8\Gamma^2} - \frac{\omega_{m0}^2 V_{m0}}{8i\Gamma^3}, \tag{B.7a}$$

$$C_m^{(5)} = \sum_n \left( \frac{\omega_{n0}}{3} + \frac{\omega_{mn}}{2} \right) \frac{V_{n0} V_{mn}}{5i\Gamma^3} - \frac{\omega_{m0}^3 V_{m0}}{30\Gamma^4}. \tag{B.7b}$$

Upon substituting Eq. (B.7) into Eq. (B.4), expanding terms up to order $\lambda^5$, and separating terms into $n = 0$ and $n \neq 0$ yields

$$
\begin{aligned}
|\Psi_\lambda\rangle = &\left[ 1 + \lambda \frac{\varepsilon_0}{i\Gamma} - \lambda^2 \left( \frac{\varepsilon_0^2}{2\Gamma^2} - \frac{V_{00}}{2i\Gamma} \right) - \lambda^3 \left( \frac{\varepsilon_0^3}{6i\Gamma^3} + \frac{V_{00}\varepsilon_0}{2\Gamma^2} \right) \right. \\
&+ \lambda^4 \left( \frac{\varepsilon_0^4}{24\Gamma^4} - \sum_m \frac{|V_{m0}|^2}{8\Gamma^2} - \frac{V_{00}\varepsilon_0^2}{4i\Gamma^3} \right) \\
&+ \left. \lambda^5 \left( \frac{\varepsilon_0^5}{120i\Gamma^5} - \sum_n \frac{\omega_{n0}|V_{n0}|^2}{30i\Gamma^3} - \sum_n \frac{|V_{n0}|^2\varepsilon_0}{8i\Gamma^3} + \frac{V_{00}\varepsilon_0^3}{12\Gamma^4} \right) \right] |\chi_0\rangle \\
&+ \sum_{n\neq 0} \left[ \lambda^2 \frac{V_{n0}}{2i\Gamma} + \lambda^3 \left( \frac{\omega_{n0} V_{n0}}{3\Gamma^2} - \frac{V_{n0}\varepsilon_n}{2\Gamma^2} \right) \right. \\
&+ \left. \lambda^4 \left( -\sum_m \frac{V_{m0} V_{nm}}{8\Gamma^2} - \frac{\omega_{n0}^2 V_{n0}}{8i\Gamma^3} + \frac{\omega_{n0} V_{n0}\varepsilon_n}{3i\Gamma^3} - \frac{V_{n0}\varepsilon_n^2}{4i\Gamma^3} \right) + \dots \right] |\chi_n\rangle.
\end{aligned}
\tag{B.8}
$$

Thus, we obtain the following inner products,

$$
\begin{aligned}
\langle\chi_0|\Psi_\lambda\rangle = &1 + \lambda \frac{\varepsilon_0}{i\Gamma} - \lambda^2 \left( \frac{\varepsilon_0^2}{2\Gamma^2} - \frac{V_{00}}{2i\Gamma} \right) - \lambda^3 \left( \frac{\varepsilon_0^3}{6i\Gamma^3} + \frac{V_{00}\varepsilon_0}{2\Gamma^2} \right) \\
&+ \lambda^4 \left( \frac{\varepsilon_0^4}{24\Gamma^4} - \sum_m \frac{|V_{m0}|^2}{8\Gamma^2} - \frac{V_{00}\varepsilon_0^2}{4i\Gamma^3} \right) \\
&+ \lambda^5 \left( \frac{\varepsilon_0^5}{120i\Gamma^5} - \sum_m \frac{\omega_{m0}|V_{m0}|^2}{30i\Gamma^3} - \sum_m \frac{|V_{m0}|^2\varepsilon_0}{8i\Gamma^3} + \frac{V_{00}\varepsilon_0^3}{12\Gamma^4} \right) + \dots,
\end{aligned}
\tag{B.9a}
$$

$$
\begin{aligned}
\langle\chi_{n\neq 0}|\Psi_\lambda\rangle = &\lambda^2 \frac{V_{n0}}{2i\Gamma} + \lambda^3 \left( \frac{\omega_{n0} V_{n0}}{3\Gamma^2} - \frac{V_{n0}\varepsilon_n}{2\Gamma^2} \right) \\
&+ \lambda^4 \left( -\sum_m \frac{V_{m0} V_{nm}}{8\Gamma^2} - \frac{\omega_{n0}^2 V_{n0}}{8i\Gamma^3} + \frac{\omega_{n0} V_{n0}\varepsilon_n}{3i\Gamma^3} - \frac{V_{n0}\varepsilon_n^2}{4i\Gamma^3} \right) + \dots
\end{aligned}
\tag{B.9b}
$$

## B.3 Various overlaps in perturbative expansion

We are ready to compute various overlaps using Eqs. (B.2) and (B.9). First, the ground state overlap $\mathcal{C}(\lambda)$ follows from Eq. (B.2a),

$$
\begin{aligned}
\mathcal{C}(\lambda) = 1 - \lambda^2 \sum_{n \neq 0} \frac{|V_{n0}|^2}{(\varepsilon_0 - \varepsilon_n)^2} \\
+ 2\lambda^3 \left( \sum_{n \neq 0} \frac{V_{00}|V_{n0}|^2}{(\varepsilon_0 - \varepsilon_n)^3} - \sum_{n \neq 0} \frac{1}{(\varepsilon_0 - \varepsilon_n)^2} \sum_{m \neq 0} \frac{\Re(V_{nm}^* V_{m0}^* V_{n0})}{\varepsilon_0 - \varepsilon_m} \right) + \dots
\end{aligned}
\tag{B.10}
$$

Second, the adiabatic fidelity $\mathcal{F}(\lambda)$ follows from Eqs. (B.2) and (B.9),

$$
\mathcal{F}(\lambda) = |\langle \Phi_\lambda | \Psi_\lambda \rangle|^2 = \left| \langle \Phi_\lambda | \Phi_0 \rangle \langle \Phi_0 | \Psi_\lambda \rangle + \sum_{n \neq 0} \underbrace{\langle \Phi_\lambda | \chi_n \rangle \langle \chi_n | \Psi_\lambda \rangle}_{=\mathcal{O}(\lambda^3)} \right|^2
$$

$$
= 1 - \lambda^2 \sum_{n \neq 0} \frac{|V_{n0}|^2}{(\varepsilon_0 - \varepsilon_n)^2}
\tag{B.11}
$$

$$
+ 2\lambda^3 \left( \sum_{n \neq 0} \frac{V_{00}|V_{n0}|^2}{(\varepsilon_0 - \varepsilon_n)^3} - \sum_{n \neq 0} \frac{1}{(\varepsilon_0 - \varepsilon_n)^2} \sum_{m \neq 0} \frac{\Re(V_{nm}^* V_{m0}^* V_{n0})}{\varepsilon_0 - \varepsilon_m} - \frac{V_{00}\varepsilon_0}{2\Gamma^2} \right) + \dots,
$$

which is identical to $\mathcal{C}(\lambda)$ (B.10) for up to order $\lambda^2$. Their difference, $\mathcal{F}(\lambda) - \mathcal{C}(\lambda)$, reads $\mathcal{F}(\lambda) - \mathcal{C}(\lambda) = -\lambda^3 \frac{V_{00}\varepsilon_0}{\Gamma^2} + \dots$

Third, $\cos^2 \theta(\lambda)$ follows from Eq. (B.9a),

$$
\cos^2 \theta(\lambda) = |\langle \Phi_0 | \Psi_\lambda \rangle|^2 = 1 - \frac{\lambda^4}{4} \sum_{n \neq 0} \frac{|V_{n0}|^2}{\Gamma^2} + \lambda^5 \frac{V_{00}\varepsilon_0^3}{\Gamma^4} + \dots
\tag{B.12}
$$

It then follows that $\sin^2 \theta(\lambda)$ and $\sin \theta(\lambda)$ read

$$
\sin^2 \theta(\lambda) = 1 - \cos^2 \theta(\lambda) = \frac{\lambda^4}{4} \sum_{n \neq 0} \frac{|V_{n0}|^2}{\Gamma^2} - \lambda^5 \frac{V_{00}\varepsilon_0^3}{\Gamma^4} + \dots,
\tag{B.13a}
$$

$$
\sin \theta(\lambda) = \frac{\lambda^2}{2} \left( \sum_{n \neq 0} \frac{|V_{n0}|^2}{\Gamma^2} \right)^{1/2} - \lambda^3 \left( \sum_{n \neq 0} \frac{|V_{n0}|^2}{\Gamma^2} \right)^{-1/2} \frac{V_{00}\varepsilon_0^3}{\Gamma^4} + \dots
\tag{B.13b}
$$

Fourth, the overlap $\sqrt{\mathcal{D}_{\mathrm{un}}(\lambda)}$ follows from Eqs. (B.2b) and (B.9),

$$
\mathcal{D}_{\mathrm{un}}(\lambda) = |\langle \Phi_\lambda | (\mathbb{I} - P) | \Psi_\lambda \rangle|^2 = \left| \sum_{n \neq 0} \langle \Psi_\lambda | \chi_n \rangle \langle \chi_n | \Phi_\lambda \rangle \right|^2 =: \lambda^6 a_6 + \lambda^7 a_7 + \dots,
\tag{B.14a}
$$

where

$$
a_6 := \frac{1}{4\Gamma^2} \left( \sum_{n \neq 0} \frac{|V_{n0}|^2}{\varepsilon_0 - \varepsilon_n} \right)^2,
\tag{B.14b}
$$

$$
a_7 := \frac{1}{2\Gamma^2} \left( \sum_{n \neq 0} \frac{|V_{n0}|^2}{\varepsilon_0 - \varepsilon_n} \right) \sum_{n \neq 0} \frac{1}{\varepsilon_0 - \varepsilon_n} \left( \sum_{m \neq 0} \frac{\Re\left( V_{nm}^* V_{m0}^* V_{n0} \right)}{\varepsilon_0 - \varepsilon_m} - \frac{V_{00}|V_{n0}|^2}{\varepsilon_0 - \varepsilon_n} \right).
\tag{B.14c}
$$

Finally, we calculate $\sin^2 \theta(\lambda)(1 - \mathcal{C}(\lambda))$ using Eqs. (B.10) and (B.13a):

$$
\sin^2 \theta(\lambda)(1 - \mathcal{C}(\lambda)) =: \lambda^6 b_6 + \lambda^7 b_7 + \dots,
\tag{B.15a}
$$

where

$$b_6 := \frac{1}{4}\left(\sum_{n\neq 0}\frac{|V_{n0}|^2}{\Gamma^2}\right)\left(\sum_{n\neq 0}\frac{|V_{n0}|^2}{(\varepsilon_0-\varepsilon_n)^2}\right),\tag{B.15b}$$

$$b_7 := -\frac{V_{00}\varepsilon_0^3}{\Gamma^4}\sum_{n\neq 0}\frac{|V_{n0}|^2}{(\varepsilon_0-\varepsilon_n)^2}$$
$$-\frac{1}{2}\left(\sum_{n\neq 0}\frac{|V_{n0}|^2}{\Gamma^2}\right)\left(\sum_{n\neq 0}\frac{V_{00}|V_{n0}|^2}{(\varepsilon_0-\varepsilon_n)^3}-\sum_{n\neq 0}\frac{1}{(\varepsilon_0-\varepsilon_n)^2}\sum_{m\neq 0}\frac{\Re(V_{nm}^*V_{m0}^*V_{n0})}{\varepsilon_0-\varepsilon_m}\right).\tag{B.15c}$$

Combing Eq. (B.15) with Eq. (B.14) and Eq. (16) yields

$$\mathcal{D}(\lambda)=\frac{\left(\sum_{n\neq 0}\frac{|V_{n0}|^2}{\varepsilon_0-\varepsilon_n}\right)^2}{\left(\sum_{n\neq 0}|V_{n0}|^2\sum_{m\neq 0}\frac{|V_{m0}|^2}{(\varepsilon_0-\varepsilon_m)^2}\right)}+16\lambda\left(\sum_{n\neq 0}\frac{|V_{n0}|^2}{\Gamma^2}\sum_{m\neq 0}\frac{|V_{m0}|^2}{(\varepsilon_0-\varepsilon_m)^2}\right)^{-2}(a_7 b_6-a_6 b_7)+\dots\tag{B.16}$$

## C  Non-interacting Hamiltonians

We consider non-interacting systems whose Hamiltonian can be written as $N$-commuting pieces in momentum space, i.e., $H_\lambda=\bigoplus_{k=1}^N\mathcal{H}_\lambda(k)$. Correspondingly, both the instantaneous ground state $|\Phi_\lambda\rangle$ and the time-evolved state $|\Psi_\lambda\rangle$ can be written as a tensor product form,

$$|\Phi_\lambda\rangle=\bigotimes_{k=1}^N|\phi_\lambda(k)\rangle,\qquad\text{and}\qquad|\Psi_\lambda\rangle=\bigotimes_{k=1}^N|\psi_\lambda(k)\rangle,\tag{C.1}$$

where $|\phi_\lambda(k)\rangle$ is the instantaneous ground state of $\mathcal{H}_\lambda(k)$, whereas for each $k$, $|\psi_\lambda(k)\rangle$ solves

$$i\Gamma\partial_\lambda|\psi_\lambda(k)\rangle=\mathcal{H}_\lambda(k)|\psi_\lambda(k)\rangle,\qquad|\psi_0(k)\rangle=|\phi_0(k)\rangle.\tag{C.2}$$

It then follows that the overlaps of various many-body wavefunctions can be written as products of overlaps of single-body wavefunctions

$$\langle\Psi_\lambda|\Phi_\lambda\rangle=\prod_k\langle\psi_\lambda(k)|\phi_\lambda(k)\rangle,\qquad\langle\Phi_0|\Phi_\lambda\rangle=\prod_k\langle\phi_0(k)|\phi_\lambda(k)\rangle.\tag{C.3}$$

Define the single-body projector $p_k:=|\phi_0(k)\rangle\langle\phi_0(k)|$ and its complementary projector $q_k:=\mathbb{I}_k-p_k$, and make use of Eqs. (C.3), we can express $\sqrt{\mathcal{D}_{\text{un}}(\lambda)}$ (15) as follows

$$\sqrt{\mathcal{D}_{\text{un}}(\lambda)}=\left|\langle\Psi_\lambda|P|\Phi_\lambda\rangle\right|\left|\prod_k(1+A_k)-1\right|,\quad\text{where}\quad A_k:=\frac{\langle\psi_\lambda(k)|q_k|\phi_\lambda(k)\rangle}{\langle\psi_\lambda(k)|p_k|\phi_\lambda(k)\rangle}.\tag{C.4}$$

To make further progress, a crucial observation for $A_k$ is that, for each $k$, the following condition holds

$$|\langle\psi_\lambda(k)|q_k|\phi_\lambda(k)\rangle|\ll|\langle\psi_\lambda(k)|p_k|\phi_\lambda(k)\rangle|.\tag{C.5}$$

This fact can be verified directly by considering a perturbative expansion in $\lambda$ similar to what has been done in App. B. If so, the following approximation formula,

$$\prod_k(1+A_k)\simeq 1+\sum_k A_k,\qquad\text{for all }|A_k|\ll 1,\tag{C.6}$$

can be applied to Eq. (C.4). Upon using Eqs. (12) and (C.6), Eq. (C.4) reads

$$\sqrt{\mathcal{D}_{\text{un}}(\lambda)}\simeq\cos\theta(\lambda)\sqrt{\mathcal{C}(\lambda)}\left|\sum_k A_k\right|.\tag{C.7}$$

**Driven Rice-Mele model**

We now apply the formalism developed above to the Rice-Mele model (32). Upon performing a Fourier transform, the Rice-Mele Hamiltonian (32) can be written as a sum of $N$ commuting terms

$$H_{\mathrm{RM}} = \sum_k \begin{pmatrix} a_k^\dagger & b_k^\dagger \end{pmatrix} \mathcal{H}_\lambda(k) \begin{pmatrix} a_k \\ b_k \end{pmatrix}, \tag{C.8a}$$

where $\mathcal{H}_\lambda(k) = \boldsymbol{d}_\lambda(k) \cdot \boldsymbol{\sigma}$ and

$$\boldsymbol{d}_\lambda(k) := \begin{pmatrix} -(J+U)-(J-U)\cos k \\ (J-U)\sin k \\ \mu(\lambda) \end{pmatrix}, \tag{C.8b}$$

with $\boldsymbol{\sigma}$ are the Pauli matrices.

We shall specialize to the case in which $J = U = \text{constant}$ and $\mu(\lambda) = \lambda$. It then follows that the $\boldsymbol{d}$ vector (C.8b) has no momentum dependence and each single-body Hamiltonian $\mathcal{H}_\lambda(k)$ (C.8b) is simply the Landau-Zener model. For this case, the $\left| \sum_k A_k \right|$ term in Eq. (C.7) simplifies

$$\left| \sum_k A_k \right| = N \frac{\sqrt{1-\left|\langle\psi_\lambda|\phi_0\rangle\right|^2}\sqrt{1-\left|\langle\phi_\lambda|\phi_0\rangle\right|^2}}{\left|\langle\psi_\lambda|\phi_0\rangle\right|\left|\langle\phi_\lambda|\phi_0\rangle\right|}, \tag{C.9}$$

where each overlap of single-body states can be obtained easily

$$|\langle\phi_\lambda|\phi_0\rangle| = |\langle\Phi_\lambda|\Phi_0\rangle|^{\frac{1}{N}} = \left(\sqrt{\mathcal{C}(\lambda)}\right)^{\frac{1}{N}}, \qquad |\langle\psi_\lambda|\phi_0\rangle| = |\langle\Psi_\lambda|\Phi_0\rangle|^{\frac{1}{N}} = (\cos\theta(\lambda))^{\frac{1}{N}}. \tag{C.10}$$

Using these results, Eq. (C.9) can be expressed in terms of $\theta(\lambda)$ and $\mathcal{C}(\lambda)$ as

$$\left| \sum_k A_k \right| = N \frac{\sqrt{1-(\cos^2\theta(\lambda))^{\frac{1}{N}}}\sqrt{1-(\mathcal{C}(\lambda))^{\frac{1}{N}}}}{(\cos\theta(\lambda))^{\frac{1}{N}}\left(\sqrt{\mathcal{C}(\lambda)}\right)^{\frac{1}{N}}} \geq \sqrt{N} \frac{\sin\theta(\lambda)\sqrt{1-(\mathcal{C}(\lambda))^{\frac{1}{N}}}}{(\cos\theta(\lambda))^{\frac{1}{N}}\left(\sqrt{\mathcal{C}(\lambda)}\right)^{\frac{1}{N}}}, \tag{C.11}$$

where we have used the inequality $(1-x)^n \leq (1-nx)$ for $0 \leq x \leq 1$ and $0 < n < 1$. It then follows that $\sqrt{\mathcal{D}_{\mathrm{un}}(\lambda)}$ (C.7) reads

$$\sqrt{\mathcal{D}_{\mathrm{un}}(\lambda)} \geq (\cos\theta(\lambda))^{1-\frac{1}{N}}\left(\sqrt{\mathcal{C}(\lambda)}\right)^{1-\frac{1}{N}}\sin\theta(\lambda)\alpha(\lambda), \tag{C.12a}$$

$$\alpha(\lambda) := \sqrt{N}\sqrt{1-(\mathcal{C}(\lambda))^{\frac{1}{N}}}, \tag{C.12b}$$

where the equality in Eq. (C.12a) holds if $\theta(\lambda)$ is small. Note that the exponent $1-\frac{1}{N}$ in Eq. (C.12a) may be approximated as 1 if $N$ is large.

# D Derivation of Eq. (42)

Combing triangle inequality (30) and the inequality $1 - \sqrt{\mathcal{F}(\lambda)} \leq \epsilon$ with $\epsilon \in [0,1]$ from quantum adiabatic theorem, we obtain

$$\begin{aligned}
\left(1 - \epsilon - \sqrt{\mathcal{C}(\lambda)}\right)^2 &\leq \mathcal{D}_{\mathrm{un}}(\lambda) \\
&\overset{(16)}{=} \sin\theta(\lambda)\sqrt{1-\mathcal{C}(\lambda)}\sqrt{\mathcal{D}(\lambda)} \\
&\overset{(39)}{=} \sin\theta(\lambda)\sqrt{1-\mathcal{C}(\lambda)}\sqrt{\mathcal{C}(\lambda)^s} \\
&\overset{(8)}{\leq} \sin\widetilde{\mathcal{R}}(\lambda)\sqrt{1-\mathcal{C}(\lambda)}\sqrt{\mathcal{C}(\lambda)^s},
\end{aligned} \tag{D.1}$$

where $C(\lambda) = e^{-C_N \lambda^2}$ as $N \to \infty$. We shall take $\lambda = \lambda_* = C_N^{-1/2}$ in the inequality above. Since we are interested in the limit where $\delta V_N / C_N \to 0$ as $N \to \infty$, we may approximate $\sin \widetilde{\mathcal{R}}(\lambda_*)$ by $\mathcal{R}(\lambda_*) = \delta V_N / (2\Gamma C_N)$ from Eq. (10),

$$
\begin{aligned}
\left(1 - \epsilon - \sqrt{\mathcal{C}(\lambda_*)}\right)^2 &\leq \sin \widetilde{\mathcal{R}}(\lambda_*) \sqrt{1 - \mathcal{C}(\lambda_*)} \sqrt{\mathcal{C}(\lambda_*)^{s_*}} \\
&\lesssim \mathcal{R}(\lambda_*) \sqrt{1 - e^{-1}} e^{-s_*/2},
\end{aligned}
\tag{D.2}
$$

where $s_* := s(\lambda_*) = -\ln \mathcal{D}(\lambda_*)$. Equation (D.2) implies

$$
\Gamma \leq \frac{1}{2} \frac{\delta V_N}{C_N} \frac{1}{\left(1 - \epsilon - e^{-1/2}\right)^2} M(s_*),
\tag{D.3}
$$

where $M(s_*) := \sqrt{1 - e^{-1}} e^{-s_*/2}$.

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
