# Peer review of "Quantum adiabaticity in many-body systems and almost-orthogonality in complementary subspace"

_SciPost Physics Core, doi:SciPost Phys. Core 8, 084 (2025)_

## Round 1 · Referee Report · Anonymous (Referee 1) · 2025-1-31

Strengths

1- pedagical presentation 2- clear writing 3- combination of analytical results and numerical examples

Weaknesses

1- unclear motivation 2- lack of discussion of relevant literature

Report

In Quantum adiabaticity in many-body systems and almost-orthogonality in complementary subspace, the authors compare adiabatic fidelity (overlap of time evolved state with the instantaneous ground state) with the overlap of the initial (ground) state with the instantaneous ground state. Bulding on previous work in Ref. [38, 39], in which the authors and collaborators noticed the effect, they set out to explain it in the present manuscript.

Overall, the manuscript takes a somewhat pedagogical approach, explaining step by step the ingredients needed for their result. I think this approach is fine, but it was difficult to me to see what in the end the precise statement is that the authors want to make. In my personal opinion, technical papers like the present one could be improved by having a "Summary and main result" Section II, in which the main assumptions are outlined and the main result is presented.

In a sense, the result is surprising: if the adiabatical fidelity and the overlaps of the instantaneous ground states are similar, then either (i) both are large, and nothing has happened (because the ground state is still essentially the same as at t=0), or (ii) both are small (the case the authors are interested in, in which case adiabatic preparation has failed, because the adiabatic fidelity is zero. Neither regime really can be called adiabatic state preparation. If adiabatic state preparation succeeds, then adiabatic fidelity should be large, and the overlap with the initial ground state small.

The difference seems to be that the authors consider a linear ramp (Eq. (7)) with a constant driving rate. Typically, when adiabatic preparation is studied, the interpolation between initial and final Hamiltonian is smooth (e.g. Gevrey class or approximations thereof) and the ramp speed is scaled down with system size, to ensure high adiabatic fidelity. There is a large body of mathematical results many-body adiabatic state preparation that the authors seem to disregard. It would be useful to understand their result in the light of what is already rigorously known about adiabatic state preparation:

[1] Sabine Jansen, Mary-Beth Ruskai, and Ruedi Seiler. “Bounds for the Adiabatic Approximation with Applications to Quantum Computation”. In: J. Math. Phys. 48.10 (Mar. 2007), p. 102111. arXiv:0603175[quant-ph].
[2] G. Nenciu. “Linear Adiabatic Theory. Exponential Estimates”. In: Commun. Math. Phys. 152.3 (1993), pp. 479–496.
[3] George A. Hagedorn and Alain Joye. “Elementary Exponential Error Estimates for the Adiabatic Approximation”. In: Journal of Mathematical Analysis and Applications 267.1 (Mar. 2002), pp. 235–246.
[4] Yimin Ge, Andr´as Moln´ar, and J. Ignacio Cirac. “Rapid Adiabatic Preparation of Injective Projected Entangled Pair States and Gibbs States”. In: Phys. Rev. Lett. 116.8 (Feb. 2016). arXiv: 1508.00570.
[5] Sven Bachmann, Wojciech De Roeck, and Martin Fraas. “Adiabatic Theorem for Quantum Spin Systems”. In: Physical Review Letters 119.6 (Aug. 11, 2017), p. 060201. arXiv: 1612.01505.
[6] Sven Bachmann, Wojciech De Roeck, and Martin Fraas. “The Adiabatic Theorem and Linear Response Theory for Extended Quantum Systems”. In: Communications in Mathematical Physics 361.3 (Aug.2018), pp. 997–1027. arXiv: 1705.02838.
[7] Sven Bachmann, Wojciech De Roeck, and Martin Fraas. The Adiabatic Theorem in a Quantum Many-Body Setting. Mar. 18, 2019. arXiv: 1808.09985.

I did not spot major omissions in the manuscript, but I am sceptical that the problem studied is really a "long-standing research stumbling block", because the studied regime is not so relevant for adiabatic state preparation and I am thus not convinced by the motivation underlying the research. As a result, I recommend publication in a more specialized venue, for example SciPost Core.

Requested changes

1- Discussion of literature 2- Clearer presentation of the main result & setting

Recommendation

Accept in alternative Journal (see Report)

  • validity: high
  • significance: low
  • originality: good
  • clarity: high
  • formatting: excellent
  • grammar: excellent

Author:  Jyong-Hao Chen  on 2025-09-14  [id 5814]

(in reply to Report 1 on 2025-01-31)
Disclosure of Generative AI use

The comment author discloses that the following generative AI tools have been used in the preparation of this comment:

OpenAI ChatGPT (GPT-5) was used to edit wording for English fluency in the reply-to-referee letter (grammar, clarity, and concision only). It was not used to generate scientific content; all edits were human-verified.

Category:
answer to question

Please refer to the attachment.

Attachment:

reply_to_referee_20250913.pdf

---

## Round 1 · Referee Report · Anonymous (Referee 2) · 2025-5-2

Strengths

1- Pedagogical tone 2- Few typos/grammatical mistake

Weaknesses

1- Unclarity of the significance of the results 2- Some confusing statements

Report

This manuscript discusses the speed-limit approach to quantum adiabaticity in quantum many-body systems.
After the introduction, the authors introduce their setup in Sec. 2, introducing the results in [37,38].
In Sec. 3, they discuss when the orthogonality limit (19) holds true, which sets the tighter bound for the difference between $\mathcal{F}$ and $\mathcal{C}$.
In particular, this limit is achieved either by (i) perturbatively considering small $\lambda$ or (ii) assuming the almost orthogonality of states for large $\lambda$.
In Sec. 4, different types of bounds, e.g., inequality (29), are obtained.
In Sec. 5, the authors verify their inequalities using the non-interacting fermionic systems.
In Sec. 6, they discuss the difference in the rates of the decrease for the quantities in their bound.
In Sec. 7, another example of an interacting Hamiltonian is discussed.
In Sec. 8, they summarize the results and state some outlooks.

In my opinion, while this paper has some new results, it is not suitable for publication in SciPost Physics.
The main reason for this evaluation is that the obtained results are not significant enough to warrant SciPost.
In my view, this manuscript mainly tries to resolve a question raised in Ref. [37], which is considered to treat a rather specific problem.
Moreover, while the almost-orthogonality provided in this manuscript is newly applied to this problem, I do not find its significance in understanding the problem of adiabatic preparation.
Indeed, for large $\lambda$, $\mathcal{F}$ quickly decays, meaning that the adiabatic preparation breaks down (in agreement with the first referee's opinion).
I also find it difficult to see what is the most important result in this manuscript in the current presentation.

In conclusion, I think that this manuscript is not suited for SciPost Physics.
After all of the questions and comments given below are sufficiently addressed, I would recommend the manuscript for publication in SciPost Physics Core.

*I think that the inequality to bound $|\mathcal{F}-\mathcal{C}|$ with $\theta$, obtained in [37], should be explicitly written down around Eq. (6).

*After Eq. (18), the authors state that it remains unclear why the values of the adiabatic fidelity and the ground state overlap are nearly identical when the system size $N$ is sufficiently large (e.g., $N \geq 100$). However, in the conclusion, they state that "we demonstrated that these refined estimates perform well even for system sizes as small as $N = O(10^2)$, ... . These results distinctly outperform the previous estimates from Refs. [37,38], which are reliable only for system sizes no smaller than $N = O(10^3)$."
I think these statements are inconsistent.

*The place of Fig. 2 could be the same as where it is referred to in the main text.

*In Fig. 2, I cannot see how the bound (e.g., (29)) is good for a large $\lambda$ region because both sides are almost zero. I think one could consider a semi-log plot to see whether (29) really offers a good bound.

*In the first sentence in Sec. 5, what is the prime in $\mathcal{D}_\mathrm{un}$?

*There is an extra space at the end of the sentence after Eq. (32c).

*Could you elaborate on the relations about $\sqrt{\mathcal{D}_\mathrm{un}}$ in the sentence after Eq. (33)? I was not able to follow the discussion at first sight.

*I am confused about how the results obtained by the authors are better than the previous ones. In Sec. 5, the authors present the bounds based on $g$ (Eq. (35)) and $f$ (Eq. (37)), which they say come from Eqs. (16) and (18), respectively. While Eq. (16) is the previously known bound, Eq. (35) seems to be better than Eq. (37), questioning the advantage of the new inequality (18). Am I misunderstanding something?

Requested changes

1- Please clearly write the motivation, their main result, and its significance. 2- Please consider all of the comments above and reflect them in the manuscript.

Recommendation

Accept in alternative Journal (see Report)

  • validity: high
  • significance: low
  • originality: ok
  • clarity: ok
  • formatting: good
  • grammar: excellent

Author:  Jyong-Hao Chen  on 2025-09-14  [id 5815]

(in reply to Report 2 on 2025-05-02)
Disclosure of Generative AI use

The comment author discloses that the following generative AI tools have been used in the preparation of this comment:

OpenAI ChatGPT (GPT-5) was used to edit wording for English fluency in the reply-to-referee letter (grammar, clarity, and concision only). It was not used to generate scientific content; all edits were human-verified.

Category:
answer to question

Please refer to the attachment.

Attachment:

reply_to_referee_20250913_KhIo3BE.pdf

---

## Round 2 · Author Response

We thank the referees for their careful reading and constructive comments on our manuscript.
We agree to transfer the manuscript to SciPost Physics Core.
In this revision, we have substantially improved the presentation in line with the referees’ requests.
Below we summarize the main changes.
For convenience, we provide a color-highlighted version of the manuscript showing all changes.
[link: https://www.dropbox.com/scl/fi/gbin3wzi0t0qrwf7yjksh/draft_adiabatic_closeness_20250913_color.pdf?rlkey=4qly8d38m9kj5p07ltvlory6d&st=0g0rj04t&dl=0]
A point-by-point response to each report will be submitted on the referee-report pages.
Sincerely,
Authors

---

## Round 2 · List of Changes

List of main changes
-
Introduction (Sec. 1). We added a subsection, Summary of main results and contributions, that summarizes our main findings and contributions. We also added a paragraph clarifying the relation of our work to rigorous results on many-body adiabatic state preparation. The literature pointed out by Referee 1 is now cited as Refs. [24,46–51].
-
Bound from Ref. [37]. As requested by Referee 2, the bound obtained in Ref. [37] now appears as Eq. (7).
-
Figure 2 revised. We added a new fourth row that overlays the curves shown in the second and third rows, with a logarithmic vertical axis; the original fourth row is now the fifth. The caption and main text were updated accordingly.
-
Finite-size scaling made explicit (Sec. 6). We added an unnumbered subsection titled Condition for adiabaticity breakdown revisited to make the finite-size scaling of the driving rate explicit.
-
Summary and Outlook (Sec. 8). We slightly revised the second paragraph to address a confusion raised by Referee 2.
-
Appendix A restored. We reinstated Appendix A: Alternative derivation of Eq. (17), which appears in our arXiv preprint but was omitted during journal submission. In- cluding this appendix improves self-containment and benefits readers, as the projection- operator technique—central to the present work and not used in Refs. [37,38]—is useful for extending the pure-state case treated here to mixed states.
-
Other changes are minor.

---

## Editorial Decision

published